# DO TRANSFORMERS UNDERSTAND POLYNOMIAL SIMPLIFICATION?

## ABSTRACT

Recently researchers have demonstrated that Transformers can be trained to learn symbolic tasks such as solving integration and differential equations in an end-to-end fashion. In these setups, for an input symbolic expression, the Transformer predicts the final solution in a single step. Since such tasks may consist of a sequence of logical steps, question remains whether such networks have understood and learnt individual steps to reach the solution. To take a deeper look, we consider the task of polynomial simplification. Polynomials can be written in a simple normal form as a sum of monomials which are ordered in a lexicographic order. For a polynomial which is not necessarily in this normal form, a sequence of simplification steps is applied to reach the fully simplified (i.e., in the normal form) polynomial. For this task, we describe a synthetic Polynomial dataset generation algorithm which generates polynomials with unique proof steps. Then, we conduct an extensive analysis of the Transformer's abilities to learn the polynomial simplification task along different dimensions.

## 1 INTRODUCTION

With the state-of-the-art performance of Deep Neural Nets (DNNs) in perceptual tasks, researchers have started to explore their logical reasoning capabilities, in particular within the domain of Automated Theorem Proving (ATP). In these domains (LEAN (de Moura et al., 2015), HOL Light and Mizar (miz, 2020)), many recent works (Paliwal et al., 2020; Aygün et al., 2020; Hahn et al., 2020) have shown that Graph Neural Networks (Gori et al., 2005; Veličković et al., 2018) and Transformers (Vaswani et al., 2017) can be trained to perform impressively on the theorem-proving task as part of a neuro-symbolic system.

In a related but different development, recently Lample & Charton (2019) showed that for symbolic integration and differential equations, a large amount of synthetic end-to-end examples can be generated using symbolic systems. In these tasks, the authors show that Transformer networks can be trained to produce the final solution from an input integral (or differential equation) in a single step. This points to the exciting possibility of using deep neural nets to learn end-to-end theorem provers, and can be beneficial for formal mathematics (Szegedy, 2020). However, the setup combines multiple reasoning steps in a single shot. Additionally, integration (or differential equation solving) is a complex task requiring understanding of the integral symbols, functions, variables, and the basic concepts of arithmetic. As the system in Lample & Charton (2019) is simply trained to output the top solution(s) and a corresponding confidence score(s), it is unclear what internal mechanisms enable these models to solve these problems. This lack of transparency has been noted in this context (Davis, 2019). An earlier work by Piotrowski et al. (2019) showed similar results for certain symbolic manipulation tasks and their work shares the same limitation.

In this paper we ask if instead of only producing the end-result of symbolic manipulation or integral, can we have the model produce a human-readable *proof* as well. While we do not know if these models reason in the way humans do, one way to produce proofs would be to "extract" a proof from the models of the above type by "probing" them in some mannner. The problem of unraveling the inner workings of Transformers by probing is an active area of research; however, at present our understanding is still evolving (Rogers et al., 2020). Hence taking a detour, we instead train the model to produce the full proof.

Inspired by Piotrowski et al. (2019), we explore a novel but simpler setting of polynomial simplification. We illustrate the task with an example. We begin with a polynomial which is a sum of product of factors, where each factor is again a sum of monomials (including constants), as shown below:

$$P_0 = (2 * x_2^2) * \overbrace{(\underbrace{3 * x_2^1}_{\text{term}} + 4)}^{\text{factor}} + \overbrace{(5 * x_1^2 + x_1^1 * x_2^1) * (3 * x_1^1) * (2)}^{\text{product}}, \qquad \text{/* Initial */}$$

To construct unique simplification steps, first each term in a factor is simplified. Once all factors are simplified (`facstep`); then within a product, all factors are multiplied (`mulstep`). Lastly simplified products are summed (`sumstep`).

$$
\begin{aligned}
P_0 &= \underline{(2 * x_2^2) * (3 * x_2^1 + 4)} + (5 * x_1^2 + x_1^1 * x_2^1) * (3 * x_1^1) * (2), & \text{/* \textsc{facstep} */} \\
&= (2 * x_2^2) * (3 * x_2 + 4) + \underline{(5 * x_1^2 + x_1^1 * x_2^1) * (3 * x_1^1) * (2)}, & (P_1), \text{/* \textsc{facstep} */} \\
&= \underline{(2 * x_2^2) * (3 * x_2 + 4)} + (5 * x_1^2 + x_1 * x_2) * (3 * x_1) * (2), & (P_2), \text{/* \textsc{mulstep} */} \\
&= (6 * x_2^3 + 8 * x_2^2) + \underline{(5 * x_1^2 + x_1 * x_2) * (3 * x_1) * (2)}, & (P_3), \text{/* \textsc{mulstep} */} \\
&= \underline{(6 * x_2^3 + 8 * x_2^2) + (30 * x_1^3 + 6 * x_1^2 * x_2)}, & (P_4), \text{/* \textsc{sumstep} */} \\
&= 30 * x_1^3 + 6 * x_2^3 + 6 * x_1^2 * x_2 + 8 * x_2^2. & (P_5), \text{/* \textsc{endpoint} */.}
\end{aligned}
$$

Piotrowski et al. (2019) explores the task of learning symbolic re-write of an entire expression. In contrast, in our setting, for step-wise prediction, at each step the system needs to find the candidate sub-expression and a relevant simplification type to perform the simplification. This setup resembles the traditional ATP setup where a system needs to learn and execute symbolic steps to reach a final solution. But it is simpler as for each step only one type of simplification is applicable. By *proof* for an initial polynomial ($P_0$) we mean the sequence of simplification steps ($P_1$ to $P_5$). A model trained on step-wise prediction task, can be used to generate a full proof. Essentially, we start with an initial polynomial, and recursively feed the model output to itself, till it generates the final simplified polynomial (in normal form). A *proof* is correct when all steps are correct.

In the above setting (termed COARSE), all terms in a factor are simplified at once in a `facstep`, and similarly all factors in a product are simplified at once in a `mulstep`. Additionally, we define another setting FINER, where a `facstep` involves simplification of a single term, and a `mulstep` involves multiplications of only two factors at once, illustrated below with an example (for `facstep`):

$$
\begin{aligned}
P_0 &= (5 * x_1^2 + \underline{x_1^1 * x_2^1}) * (3 * x_1^1) * (2), & \text{/* \textsc{facstep} */} \\
&= (5 * x_1^2 + x_1 * x_2) * \underline{(3 * x_1^1)} * (2), & \text{/* \textsc{facstep} */} \\
&= \underline{(5 * x_1^2 + x_1 * x_2) * (3 * x_1)} * (2).
\end{aligned}
$$

As a state-of-the-art model, we explore Transformers. While both Graph Neural Networks and Transformers have been used for single-step representation learning of symbolic theorems and single step goal-theorem scoring, Transformer-based sequence-to-sequence networks have shown superiority in end-to-end tasks in integration, differential equations (Lample & Charton, 2019) and temporal logic (Hahn et al., 2020) domains. Hence for the aforementioned tasks of step-wise polynomial simplification, we explore the Transformer's ability along several dimensions. Our contributions are the following: 1) we propose polynomial simplification tasks requiring multiple steps of symbolic manipulation, 2) we show how datasets of different configurations can be generated synthetically for the task, 3) we propose an array of metrics to dissect the performance of Transformers, and 4) lastly through extensive experiments we show the performance of the Transformer on this task, establishing a strong baseline for future endeavors.

**Results Summary** By varying over coefficient size, proof granularity and input representation (in Tables 1, 2, Appendix Table 6) we observe that 1) full proof accuracy is only slightly lower than single-shot endpoint prediction accuracy in many 1-variable configurations, 2) coarse granular proofs help learn somewhat more accurate proofs, 3) prefix representation helps in most cases but infix sometimes provides higher accuracy. More than 80% errors (Tab. 7 and 8 in Appendix) occur in multiplication steps, and we observe (through independent experiments) Transformer's struggle

to learn how to multiply numeric coefficients. By letting the system annotate the candidate sub-expression, we observe that the system can understand candidate sub-expressions and which next step to perform explicitly (Tables 3, and Appendix Tables 9, 10, 11). Also, through visualization we observe similar effects (Figures 1, 2 Appendix). We see systems trained for 2-variable outperform corresponding 1-variable systems on 1-variable test sets. For 1 variable, we observe steady and significant higher gains (till 10% for full proof) using curriculum learning (Table 17 Appendix).

## 2    RELATED WORK AND DISCUSSION

Unlike the problems dealt with by the aforementioned automatic theorem provers and related neural-based systems, polynomial simplification does not involve any search. Our problem is simpler and tests a specific ability, namely certain kinds of symbol manipulations. This simplicity affords certain advantages (shared by Piotrowski et al. (2019) and Lample & Charton (2019)): (1) We can generate artificial data to train models without limitations on the size. (2) It is easier to test the abilities of the models more thoroughly along multiple axes. (3) Accuracy achieved is much higher than for harder tasks, suggesting that fully solving such tasks may be possible in the near future.

To compare with symbolic manipulation systems we note that in more detail the ability tested by our task is the following: the model must be able to identify smallest parts of the polynomial that can be simplified: (1) simplification of a factor, (2) multiplication of two factors, (3) addition of two sub-polynomials. Having identified what simplification to apply the model must produce a new polynomial with just that simplification. This ability is not tested by previous neural-based symbolic manipulation systems such as Piotrowski et al. (2019) and Lample & Charton (2019) and related works such as Saxton et al. (2019) and Hahn et al. (2020). Several recent works have produced synthetic datasets for theorem proving tasks (Aygün et al., 2020; Wu et al., 2020; Polu & Sutskever, 2020), however, their focus remains more on search-based proofs.

## 3    POLYNOMIAL SIMPLIFICATION DATASET

We proceed similarly to Lample & Charton (2019) to generate the symbolic polynomials and simplified steps synthetically using the `Sympy` library of Python. To have a fine-grained control over the generated polynomials and well-defined proof steps, we consider polynomials which are sums of products[1]. We also note that symbolic generation using the `Sympy` library lets us ensure correctness of each generated expressions and validity of each steps.

### 3.1    NOTATIONS

We start with the set of variables $x_{\mathcal{P}} = \{x_1, \ldots, x_{\mathrm{nvar}}\}$. We represent the starting point polynomial $\mathcal{P}_0$ in $x_{\mathcal{P}}$ as the sum of products of factors:

$$\mathcal{P}_0 = P_1 + P_2 + \ldots + P_{\mathrm{nprod}},$$

$$P_i = \prod_{j=1}^{\mathrm{nfac}_i} f_{ij}, \tag{1}$$

where each factor ($f_{ij}$) has the form $f = \sum_k (a_k * \prod_l x_{kl}^{d_{kl}})$, where $x_{kl} \in x_{\mathcal{P}}$ (dropping $i, j$ for clarity). Here coefficients $a_k \in \mathbb{N}^+$, and powers of the variables $d_{kl} \in \mathbb{N}$. nprod is the number of products and $\mathrm{nfac}_i$ denotes the number of factors in $P_i$.

We denote the set of factors as $f_{\mathcal{P}} = \{f_{ij} | \exists i, P_i = \prod_{j=1}^{\mathrm{nfac}_i} f_{ij}\}$. The simplified endpoint polynomial is of the form $\hat{\mathcal{P}} = \sum_{m=1}^{q} \hat{t}_m$, where $\hat{t}_m = \hat{a}_m * \prod_n x_n^{d_{mn}}$, where $x_n \in x_{\mathcal{P}}$. We use the symbol $\hat{P}_i$ to denote the simplified form of $P_i$. The functions $\mathrm{terms}(), \mathrm{vars}(), \mathrm{coeffs}()$ returns a list of terms, variables, coefficients in the input expression. Our sampling algorithm guarantees that the generated polynomial and its simplified endpoint abides by constraints on number of terms, products, factors and variables; limit on degree and coefficient sizes. An example is $\mathrm{nprod} \in \{2, \ldots, \mathrm{maxP}_{\mathcal{P}}\}$ (The full list is provided in Appendix Table 4).

---

[1] The generation algorithm in Lample & Charton (2019) may generate nested sums and products. For such polynomials, an unique proof sequence is hard to define which makes whole *proof*s harder to evaluate. Our restriction over the form of the polynomial helps us generate unique proofs, which are easier to evaluate.

## 3.2 BUILDING A POLYNOMIAL PROOF

Here, we briefly describe the starting polynomial generation process; detailed algorithm is in the appendix. Any randomly sampled polynomial (represented as a sum of products) can be included as a starting point in the dataset as long as the polynomial respects certain configuration parameters (in Appendix Table 4). This is unlike Lample & Charton (2019), where many randomly generated integrals (or differential equations) might not have a solution. Hence, we randomly sample the constraint parameters in a top-down manner; and then construct terms, factors and products in a bottom-up manner using the parameters. We first sample the following 1) a set of participating variables ($x_\mathcal{P}$), 2) maximum degree for any monomial in the simplified polynomial (mdeg), and 3) the number of products in the starting polynomial (nprod). We then call the algorithm `buildProduct` (Algorithm 1 in appendix) to create nprod individual products.

**Building a Product**  In `buildProduct` (Algorithm 1 in Appendix), first we sample $\text{nfac}_i$, the maximum number of factors in the product ($P_i$). We then build factors sequentially. For each new factor, we sample a subset of variables in a factor. We pass on product-level constraints such as maximum degree in a product, maximum terms in a product, and maximum coefficient for a product as rdegree, rterms and rcoeff respectively; and call the sub-routine `buildFactor` (Algorithm 2 to create a factor. After a factor is sampled, the constraints rdegree, rterms and rcoeff are updated. `buildFactor` is used to create at most $\text{nfac}_i$ factors, that all abide by the above constraints and stops if the limit of maximum degree in the product is reached. The terms in a factor are arranged in a lexicographical order. Since, this sequential generation of factors may induce a certain pattern of decreasing degrees and coefficients, we shuffle the factors to create the final product.

**Simplification Steps and Full Proof**  For both COARSE and FINER configurations, we build the *proof* steps in the following way: 1) first we do a sequence of `facsteps` where terms get collected within a factor (such as $2x + 3x$ to $5x$, $x^1$ and $1x$ becomes $x$); 2) then a sequence of `mulsteps` are performed where simplified factors are multiplied out; and 3) lastly, in `sumstep` simplified products are added together. As mentioned before, the sequence of simplification steps till the endpoint constitute a full *proof*.

## 4 EXPERIMENTS

### 4.1 DATASET

We vary dataset configurations along the following dimensions:
• Number of Variables in polynomial, product and factor is varied between 1 and 2.
• Coefficients Size: Maximum coefficient in the polynomial, product and factor are gradually varied from $\{60, 20, 5\}$ (SMALL), to $\{120, 40, 8\}$ (MEDIUM) and $\{300, 100, 10\}$ (LARGE). DEFAULT is $\{120, 40, 8\}$.
• Maximum degree in polynomial and a factor has two configurations: $\{6, 3\}$ (DEFAULT), and $\{12, 5\}$ (MEDIUM DEGREE).
• Maximum number of terms in a simplified product and a factor has two configurations: $\{8, 3\}$ (DEFAULT), and $\{20, 4\}$ (MEDIUM TERMS). For the latter, we also set maximum products in a sum and maximum factors in a product as 5 and 4 respectively.
• No Backtrack: We also try a very large configuration (NO BACKTRACK) where maximum coefficients in polynomial, product and factor are $\{10125, 3375, 5\}$, maximum degree in polynomial and factor are set to $\{9, 3\}$. Maximum terms in a product is set to 27. This is a configuration, where no sampled factor, or product is ever rejected for violating any higher-level constraint.

**Infix and Prefix**  We focus on exploring seq2seq networks for all our experiments. We consider the prefix and infix traversals of the abstract syntax tree of the polynomial input as sequences. Lample & Charton (2019) briefly touched upon the usefulness of the prefix notation over infix, but do not provide any empirical evidence supporting the statement. Hence, in our experiments, we consider both INFIX and PREFIX representations.

## 4.2 TASKS AND METRICS

We identify two central tasks : 1) Step-wise prediction: where an input polynomial is provided and the task is to perform the next proof step, and 2) Endpoint Prediction: where given a polynomial, the task is to predict the fully simplified polynomial in a single step.

To compare with the Endpoint prediction task, we use the Step-wise prediction task to compute the full *proof* accuracy as the percentage of proofs where all individual proof steps are accurate[2]. Apart from the accuracy, we also compare the examples seen by the systems trained in the above two types of tasks. For the Step-wise task, a training example corresponds to an individual simplification step; whereas for the Endpoint task an example is a pair denoting the initial and the endpoint polynomial. We also report the following: 1) error percentages grouped by each different types of steps `facstep`, `mulstep`, and `sumstep`, 2) calibration scores of the systems based on a threshold. To compute accuracy for an example (in both the tasks), we use the `simplify` method of `Sympy` library and check symbolically whether the difference between the predicted expression and the ground-truth expression is equal to zero.

**Calibration**: As end-to-end models grow more accurate and their usage increases, it's important that the users can trust such models. In addition to reporting each simplified step and a confidence score, we also report calibration score computed from the ratio of the top two outputs predicted for each step (using beam width 5). Using a Calibration constant threshold (usually 5), we report the sure rate which is percentage of times when the ratio (in log scale with base e) exceeds the threshold. We also report precision, recall and F-1 score for calibration.

## 4.3 MODEL

Adapting the experimental setup by Lample & Charton (2019)[3], we train a seq2seq network to predict the next proof step provided a polynomial as a sequence. For all the experiments, we train a Transformer network (Vaswani et al., 2017) architecture with 4 attention heads, 4 encoder and decoder layers with hidden embedding size of 256. We use an Adam optimizer (Kingma & Ba, 2014) with a learning rate of $10^{-4}$. We limit the maximum token length to $512$ and use a batch size of 32 polynomial pairs.

During training, we synthetically generate each batch of equations. To avoid collisions between train and test sets, we first use a fixed seed to generate the test and the validation sets of polynomial simplification full proofs and collect the simplified end-points. We make sure that the simplified versions of the input polynomial in the training batches, do not collide with any endpoints in the the test and validation set. Authors in Piotrowski et al. (2019) shows that probability of such collisions in the generated integration dataset by Lample & Charton (2019) to be quite high, and urges to report the test accuracy by accounting for such collisions explicitly.

During inference, we use beam-search with different beam widths (beam 1 and 5) to decode the expressions. For our results, beam width 1 is used for proof accuracy. Calibration results are produced using beam 5 decoding. During decoding, if any malformed (prefix or infix) expressions are generated, we report the percentage of such expressions[4].

## 4.4 EXPERIMENT ORGANIZATION

In the next sub-sections, we provide problem space-size estimate (§4.5) to understand if the accuracies are an effect of memorization. Then we vary the proof granularity, coefficient configurations and input representation to test Transformers' accuracy and errors in both tasks (§4.6). Next, to assess whether Transformers can specifically predict candidate next sub-expression to be simplified, we try an annotated proof setting (§4.6.1). To estimate the learning ability of addition and multiplication on symbolic variables, we test a setting where the coefficients are also symbolic, thus bypassing the need for the Transformer to do integer multiplication. Next, we discuss out-of-distribution general-

---

[2]We have also attempted recursive proof generation, where the output from the decoder is fed to the encoder in the next step. It does not vary from the teacher-forcing since, if in any step the model is wrong, the model does not recover after that.

[3]`https://github.com/facebookresearch/SymbolicMathematics`

[4]Similar to Lample & Charton (2019), we find that the percentage of malformed outputs was very low ($<$ 0.5%). So we did not explicitly correct for it.

ization ability of the systems (§4.7). We also explore several curriculum strategies to take advantage of the well-defined sub-tasks and their varying complexities (§4.8). Lastly, we provide layer-wise attention visualizations of a trained system in the Appendix (Figs. 1 & 2).

## 4.5 PROBLEM SPACE SIZE ESTIMATION

For smaller configurations, it is probable that eventually all simplified polynomials would be included in the training data. To account for this, we estimate the problem space size for each configuration and report the size of training data for comparison. We randomly generate two sets of starting polynomials say $S_1$ and $S_2$, and check for collisions among them. Assuming the actual size is $X$ and uniform distribution over all starting polynomials, the expected number of collisions would be $R = \frac{S_1 * S_2}{X}$. Using the above method, we estimate the number of un-simplified polynomials and the number of unique endpoints, and report in Appendix Table 5. We observe that compared to the number of training examples it took for the models to converge in both End-point and Step-wise prediction tasks, the space of possible equations is often 25 (or more) times higher.

Sampled polynomials are not uniformly distributed as we assign an equal probability while sampling polynomials of lower and higher degrees, say 3 and 6; whereas there are more polynomials of degree 6 than degree 3. For non-uniform distributions, we expect more collisions as higher probability equations are more likely to occur in both $S_1$ and $S_2$. Moreover, since many equations may map to the same endpoint, such collisions for endpoints are even more likely. Thus, our empirical estimate of the population size provides a lower bound on the true value.

## 4.6 INPUT REPRESENTATION

We report the results for one and two variables, for all configurations in Tables 1 and 2. In Table 1, we include results for both COARSE and FINER configurations. We observe that COARSE proof-steps with PREFIX representation provides the best full proof accuracy for four out of six configurations (especially for larger coefficient sizes). Across COARSE and FINER, in five out of six configurations PREFIX representation increases the full proof accuracy over INFIX, while the improvement is not always substantial. In SMALL COEFF configuration, the FINER setting improves over COARSE for full proof accuracy. From the calibration results, we see that the winning combinations often provide the highest calibration F-1 score (more prominent for 2 variables), indicating lesser ambiguity in the decision made. In Table 2, using PREFIX representation for two variables provides 3 to 4% boosts in full proof accuracy for 4 out of 6 configurations. Since, FINER steps do not improve full proof accuracy for two variables, we report the results in Table 6 in the appendix. However, for NO BACKTRACK, the infix representation clocks a 9.5% improvement over prefix. Comparing with Endpoint accuracy, as coefficient sizes grow from SMALL to NO BACKTRACK, for 1 variable, the Endpoint accuracy is only slightly higher (1 to 2%) than the full proof accuracy. However, for MEDIUM TERMS and MEDIUM DEGREE, the Endpoint accuracy shows a 3.6% and 13% improvement respectively. For 2 variables, Endpoint task accuracy is larger in most cases.

In Tables 7 and 8 (in Appendix) we show the model errors for each step type. We observe that more than 80% of the model errors occur in the multiplication step. In the MEDIUM TERMS setting, factor simplification causes 15-25% of the errors, possibly because of higher number of factors to simplify. For 2 variable case, addition step accounts for 10-15% of the errors. In all other cases, both factor simplification and addition cause close to 5% of the model errors each. As mentioned in §4.4, we experimented with symbolic coefficients to mitigate the difficulties with integer multiplication. This however didn't give good results possibly due to output becoming too long.

### 4.6.1 ANNOTATED PROOFS

In each step, simplification is performed over a sub-expression of the polynomial. To check explicitly, if the system can locate the sub-expression and find the type of simplification step, we devise the annotated proof setting. For each simplification step, we add an intermediate step, in which the model *annotates* the part of polynomial to operate on. For example, the starting input sequence is "MARK $ $(5 * x_1^2 + x_1 * x_2) * (3 * x_1) * (2)$"; and the corresponding expected output sequence is "MUL $ #$(5 * x_1^2 + x_1 * x_2) * (3 * x_1)# * (2)$". Each sequence has two parts: 1) the step index to perform (MARK, MUL, FAC, SUM), and 2) the polynomial expression. For MARK step, a marker token (#) is used to annotate the candidate sub-expression to be simplified next.

| | | Endpoints | | #Train | Full Proof | | Step-wise | | Calibration | | | |
|---|---|---|---|---|---|---|---|---|---|---|---|---|
| | | #EE | #Endpoint Acc | | Full Proof Accuracy | Stepwise Accuracy | Top-1 Acc | Beam-5 Acc | Sure Rate | P | R | F1 |
| Small Coeff | Coarse/Infix | 5M | 96 | 3.6M | 95 | 98.83 | 88.13 | 89.67 | 83.2 | 100 | 94.4 | 0.97 |
| | Fine/Infix | 5M | 96 | 4.8M | **98.9** | **99.79** | 94.46 | 95 | 92.38 | 100 | 97.8 | **0.99** |
| | Coarse/Prefix | 5.2M | 97.8 | 3.2M | 95.3 | 98.97 | 87.83 | 89.37 | 83.03 | 100 | 94.54 | 0.97 |
| | Fine/Prefix | 5.2M | 97.8 | 4.4M | 96.9 | 99.4 | 95.1 | 95.83 | 93.13 | 99.96 | 97.9 | **0.99** |
| Medium Coeff | Coarse/Infix | 4.1M | 91.2 | 4.3M | 92.8 | 98.24 | 88.97 | 91.67 | 84.3 | 100 | 94.75 | 0.97 |
| | Fine/Infix | 4.1M | 91.2 | 2.9M | 90.3 | 97.99 | 86.1 | 87.68 | 81.14 | 100 | 94.24 | 0.97 |
| | Coarse/Prefix | 6.1M | 95.87 | 5.3M | **93.6** | **98.58** | 86.6 | 88.47 | 82.83 | 99.88 | 95.54 | **0.98** |
| | Fine/Prefix | 6.1M | 95.87 | 4.5M | 91.7 | 98.37 | 95.1 | 96.43 | 91.27 | 100 | 95.97 | **0.98** |
| Large Coeff | Coarse/Infix | 4.8M | 83.73 | 3.4M | 82.1 | 95.97 | 92.34 | 94.22 | 87.2 | 99.98 | 94.41 | 0.97 |
| | Fine/Infix | 4.8M | 83.73 | 3.4M | 82.5 | 96.44 | 92.32 | 94.26 | 87.5 | 99.98 | 94.76 | 0.97 |
| | Coarse/Prefix | 6.5M | 85.87 | 3.5M | **83.5** | **96.25** | 80.6 | 83.3 | 75 | 99.91 | 92.97 | 0.96 |
| | Fine/Prefix | 6.5M | 85.87 | 3.2M | 82 | 96.32 | 79.13 | 80.63 | 75.57 | 99.96 | 95.45 | **0.98** |
| No Backtrack | Coarse/Infix | 5.9M | 80.1 | 3.8M | 75.6 | 94.62 | 72.74 | 77.28 | 61.8 | 99.9 | 84.88 | 0.92 |
| | Fine/Infix | 5.9M | 80.1 | 4M | 74.5 | 94.76 | 88.34 | 90.9 | 79.44 | 99.92 | 89.86 | 0.95 |
| | Coarse/Prefix | 6.6M | 78.87 | 5.6M | **79.7** | **95.38** | 81.93 | 85.57 | 72.2 | 100 | 88.12 | 0.94 |
| | Fine/Prefix | 6.6M | 78.87 | 4.2M | 74.7 | 95.23 | 79 | 82.03 | 72.23 | 100 | 91.43 | **0.96** |
| Medium Degree | Coarse/Infix | 9.2M | 96.4 | 4.9M | **92.8** | **98.26** | 87.18 | 88.96 | 81.12 | 100 | 93.05 | 0.96 |
| | Fine/Infix | 9.2M | 96.4 | 3.3M | 83.4 | 96.12 | 88.26 | 90.44 | 83.04 | 99.95 | 94.04 | **0.97** |
| | Coarse/Prefix | 7M | 94.33 | 4.3M | 87.7 | 96.82 | 77.33 | 82.13 | 69.33 | 100 | 89.66 | 0.95 |
| | Fine/Prefix | 7M | 94.33 | 5.9M | 90.6 | 97.92 | 82.2 | 83.7 | 77.27 | 99.96 | 93.96 | **0.97** |
| Medium Terms | Coarse/Infix | 4.6M | 81.9 | 2.3M | 72.7 | 93.99 | 79.44 | 82.22 | 68.22 | 99.97 | 85.85 | 0.92 |
| | Fine/Infix | 4.6M | 81.9 | 2.8M | 75.1 | 95.42 | 86.2 | 88.48 | 76.72 | 99.92 | 88.93 | 0.94 |
| | Coarse/Prefix | 7M | 89.8 | 4.3M | **76.3** | **95.78** | 87.8 | 91.17 | 81.3 | 100 | 92.6 | **0.96** |
| | Fine/Prefix | 7M | 89.8 | 3.2M | 74.8 | 95.55 | 87.67 | 90.4 | 76.9 | 100 | 87.72 | 0.93 |

Table 1: Results for 1 variable in the COARSE and FINE configuration for both Infix and Prefix representation.

| | | Endpoints | | #Train | Full Proof | | Step-wise | | Calibration | | | |
|---|---|---|---|---|---|---|---|---|---|---|---|---|
| | | #EE | #Endpoint Acc | | Full Proof Accuracy | Stepwise Accuracy | Top-1 Acc | Beam-5 Acc | Sure Rate | P | R | F1 |
| Small Coeff | Infix | 4.3M | 94.7 | 3.7M | 87.9 | 97.01 | 88.9 | 91 | 81.07 | 100 | 91.19 | 0.95 |
| | Prefix | 4.5M | 93.93 | 5.3M | **91.2** | **98.08** | 83.83 | 86.7 | 77.57 | 100 | 92.52 | 0.96 |
| Medium Coeff | Infix | 7M | 95.3 | 5.3M | **88.5** | **97.35** | 90.98 | 93.7 | 84.64 | 99.98 | 93.01 | 0.96 |
| | Prefix | 5.2M | 92.77 | 4.8M | 84.5 | 96.03 | 89.57 | 92.93 | 81.27 | 99.96 | 90.7 | 0.95 |
| Large Coeff | Infix | 9M | 91.8 | 3.8M | 80.4 | 95.18 | 90.44 | 93.14 | 82.74 | 99.93 | 91.42 | 0.95 |
| | Prefix | 6.1M | 86.6 | 5.4M | **83.7** | **96.23** | 92.23 | 94.57 | 86.03 | 100 | 93.28 | 0.97 |
| No Backtrack | Infix | 8.6M | 83.8 | 5M | **72.7** | **93.13** | 75.48 | 78.74 | 64.4 | 100 | 85.32 | 0.92 |
| | Prefix | 7.1M | 79.2 | 4.3M | 63.2 | 89.87 | 72.07 | 76.43 | 59.4 | 99.94 | 82.38 | 0.9 |
| Medium Degree | Infix | 4.9M | 87.9 | 5.1M | 80.5 | 95.13 | 90.3 | 92.53 | 80.63 | 100 | 89.29 | 0.94 |
| | Prefix | 5.2M | 83.73 | 6.1M | **83.4** | **96.41** | 92.07 | 94.43 | 83.13 | 99.96 | 90.26 | 0.95 |
| Medium Terms | Infix | 8.5M | 90 | 3.8M | 64 | 92.03 | 80.5 | 83.66 | 66.62 | 100 | 82.76 | 0.91 |
| | Prefix | 6.6M | 87.07 | 6.3M | **67.8** | **93.58** | 89.7 | 91.57 | 80.33 | 99.96 | 89.52 | 0.94 |

Table 2: Results for 2 variables for the COARSE configuration for both Infix and prefix representations.

We experiment only with INFIX representation. The results for 1 variable and 2 variables are in Table 3 and 9 (in Appendix). The errors per step type are shown in Appendix Tables 10 and 11. Compared to non-annotated setting, while the step-wise accuracy is similar, the proof accuracy suffers often by 7-10%. A reason for such decrease in accuracy is that length of the annotated proofs are twice as long as non-annotated. However, we note that the errors in MARK step are the lowest compared to other types of steps. This indicates that the models are able to learn the candidate sub-expression for simplification, and predict the next operation correctly.

## 4.7 OUT-OF-DISTRIBUTION EVALUATION

We also test out-of-distribution generalization by choosing different test configurations than train. The best 2 Variable models (COARSE/PREFIX) were tested on 1 Variable dataset with same coefficient configuration. We interestingly observe (in Appendix Table 14) that in all settings except one

| Config | Proof Type | Endpoint | | #Train | Full Proof | | Stepwise | | Calibration | | | |
|---|---|---|---|---|---|---|---|---|---|---|---|---|
| | | #EE | Endpoint Acc. | | Full Proof Acc. | Greedy Stepwise Acc. | Top-1 Acc. | Beam-5 Acc. | Sure Rate | P | R | F1 |
| SMALL COEFF | Fine | 5M | 96 | 2.4M | 88.5 | 98.82 | 86.77 | 87.7 | 83.97 | 99.96 | 96.73 | 0.98 |
| | Coarse | | | 3.7M | **91.9** | **99.16** | 90.07 | 90.73 | 88.13 | 100 | 97.85 | 0.99 |
| MEDIUM COEFF | Fine | 4.1M | 91.2 | 2.8M | 78.6 | 97.66 | 92.67 | 93.63 | 88.23 | 100 | 95.22 | 0.98 |
| | Coarse | | | 3.5M | **84.2** | **98.29** | 94.83 | 95.53 | 92.4 | 99.96 | 97.4 | 0.99 |
| LARGE COEFF | Fine | 4.8M | 83.73 | 3.6M | 75.5 | 97.37 | 96.8 | 97.8 | 92.4 | 99.93 | 95.39 | 0.98 |
| | Coarse | | | 4.6M | **80.3** | **97.86** | 80.37 | 81.6 | 77.83 | 100 | 96.85 | 0.98 |
| NO BACK TRACK | Fine | 5.9M | 80.1 | 4.1M | **68** | **96.78** | 90.43 | 92.33 | 84.5 | 99.96 | 93.4 | 0.97 |
| | Coarse | | | 3.6M | 59.7 | 95 | 92.5 | 94.33 | 86.47 | 99.81 | 93.3 | 0.96 |
| MEDIUM DEG | Fine | 9.2M | 96.4 | 3.7M | 76 | 97.37 | 83.67 | 85.23 | 79.1 | 100 | 94.54 | 0.97 |
| | Coarse | | | 3.4M | **78.7** | **97.38** | 93.2 | 94.37 | 88.2 | 100 | 94.64 | 0.97 |
| MEDIUM TERMS | Fine | 4.6M | 81.9 | 3.6M | **70.4** | **97.48** | 91.5 | 92.2 | 86.87 | 100 | 94.94 | 0.97 |
| | Coarse | | | 3.3M | 66.2 | 96.34 | 88.9 | 90.27 | 83.17 | 99.84 | 93.4 | 0.97 |

Table 3: Results for FINE and COARSE configurations for 1 Variable for annotated proofs

(MEDIUM COEFF), the 2 variable models outperform the corresponding 1 variable models. For the LARGE COEFF case, the improvement is close to 6% over the 1 variable model. As expected, the 2 Variable models perform better on 1 variable dataset than 2 variable. The results for OOD evaluation with respect to coefficient limits, polynomial degree and polynomial length (no. of terms in starting polynomial) are discussed in the Appendix (Tables 15 & 16).

## 4.8 CURRICULUM LEARNING

Simplification steps entail learning of addition and multiplication of numeric coefficients and symbolic variables. But, as some of the individual sub-tasks seem harder to grasp, we employ different types of curricula based on the Mastering-Rate-based (MR) curriculum learning algorithm proposed by Willems et al. (2020)[5]. For all our experiments, we use the MR algorithm with gAmax Linreg A2D converter functions described in Willems et al. (2020). Model parameters and the training configurations remain the same. We show the results in Table 17 for 1 variable COARSE configuration. As coefficient size grows from SMALL, MEDIUM, LARGE to NO BACKTRACK, improvements in full proof accuracy steadily increase from 1% to 10.84% (COARSE/INFIX). For NO BACKTRACK, the improvement in top-1 accuracy is by 20% from a no curriculum setting. However, we observe for MEDIUM TERMS, there is a drop in accuracy for all curricula and input representations. It is possible that, more carefully designed curricula may improve the results. There is no clear advantage observed between infix or prefix representations. However, compared to learning without a curriculum, the improvement observed for infix representation is often larger than prefix.

## 5 CONCLUSION

We explored the polynomial simplification task to investigate the capabilities and shortcomings of Transformer networks across various dimensions. We proposed a synthetic polynomial generation algorithm which generates constrained polynomials with unique proof steps. While Transformers perform impressively in many settings, reaching above 90% proof accuracies, there were also clear limitations and there are many avenues for future work. Among notable results, in many cases full proof accuracy is lower than endpoint accuracy, but with a low margin. This is perhaps not surprising because the model is trained to optimize for stepwise accuracy and generating a valid proof requires getting all of the multiple proof steps correct. Thus a more proof-centric training approach might further improve proof-wise accuracies. Prefix representation has a slight advantage over infix and coarse proofs have slight advantage over fine proofs. Transformers quickly learn addition, but consistently struggle with multiplication. Carefully designed curriculums can boost full proof accuracy up to 10% for large coefficient sizes. Models trained on two variable datasets often did very well on single variable datasets—even better than the models trained on single variable datasets. Exploring multivariate polynomial manipulations and more general algebraic systems are some immediate future directions, though even for the polynomial simplification task significant gaps remain in our understanding.

---

[5]For full details, please see Appendix Section I.

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

## A  VISUALIZATION OF TRANSFORMER LAYERS

Following Piotrowski et al. (2019), we also attempt to understand what the Transformer networks are learning through layer-wise visualization of attention (Vig, 2019). We take model trained on COARSE granularity proofs using INFIX representation for 1 variable using the SMALL COEFF configuration. We take the following example:

$$P_0 = \underline{(4 * x_1^2) * (5 * x_1^3 + 4 * x1)} + (12 * x_1), \qquad \text{/* MULSTEP */}$$
$$= (20 * x_1^5 + 16 * x_1 ** 3) + (12 * x_1)$$

In Figure 1, we observe that in layer 2 encoder-decoder attention indicates that while generating the number 16, the Transformer network is clearly able to attend to the two digits 4 and 4 required for the multiplication. In Figure 2, we observe that the Transformer networks, in the same time also learns to copy the expression $12 + x_1$ in Layer 1. Even though such clear logical patterns emerge quite frequently, in some cases patterns become hard to interpret.

---

**Algorithm 1:** BuildProduct (Sampling Products)

---

**Input:** $x_{\mathcal{P}}$, mdeg
**Constraints:** nvars_prod, max_coeff_prod, max_fac_prod, max_terms_prod
**Output:** A list of factors $F_{seq}$

1 Sample $nvar \in \{\text{num\_vars\_fac}, \ldots, \text{nvars\_prod}\}$
2 $nvar = min(|x_{\mathcal{P}}|, nvar)$
3 Sample $nvar$ variables from $x_{\mathcal{P}}$ as $x_{\mathcal{P}_i}$         // Variable set for this product
4 Sample $nfac \in \{2, \ldots, \text{max\_fac\_prod}\}$          // #Factors for this product
   /* Get maximum degree, terms and coefficient available */
5  $rdegree = mdeg, rterms = \text{max\_terms\_prod}, rcoeff = \text{max\_coeff\_prod}$
6  $cprod = 1$                // Keeping track of product built till now
7  $F_{seq} = [\,]$
8 **for** $j \leftarrow 1$ **to** $nfac\ 1$ **do**
9   | $f_j = \text{buildFactor}(x_{\mathcal{P}_i}, rdegree, rterms, rcoeff)$
    |   /* Update degree, terms and coefficient for next factor */
10  | $cprod = cprod * f_j$
11  | $rdegree = rdegree - degree(f_j)$
12  | $rterms = \text{max\_terms\_prod}/|terms(cprod)|$
13  | $rcoeff = \text{max\_coeff\_prod}/max(coeffs(cprod))$
14  | Append $f_j$ in $F_{seq}$
15  | **if** $rdegree == 0$ **then**
16  |   | break
17  | **end**
18 **end**
19 Shuffle $F_{seq}$

---

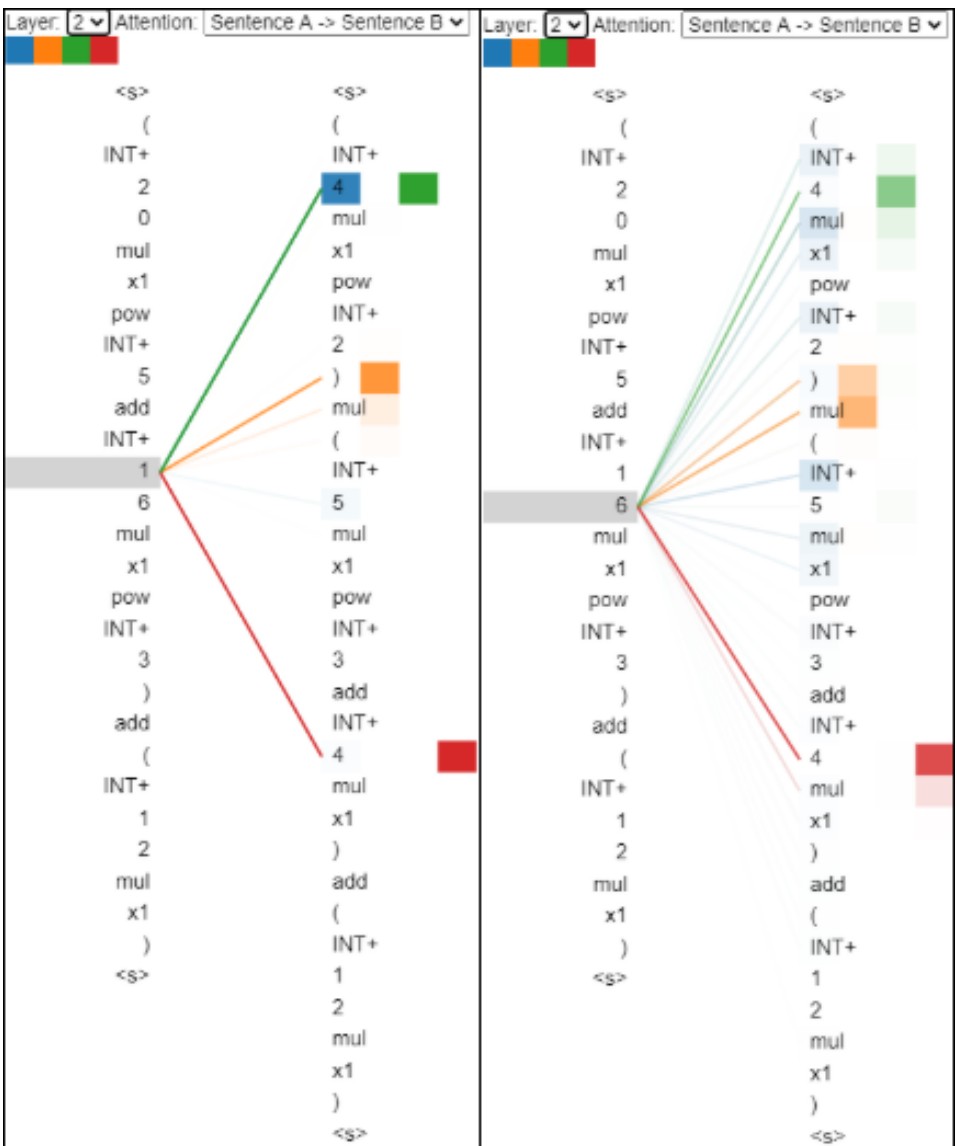

Figure 1: The layer 2 encoder-decoder attention for the output digits 16 in the first simplified product for the output $(20 * x_1^5 + 16 * x_1 * *3) + (12 * x_1)$. As expected, the digits 1 and 6 attends to the coefficients of the first and third monomial in the input expression $\underline{(4 * x_1^2) * (5 * x_1^3 + 4 * x1)} + (12 * x_1)$. Config: COARSE, SMALL COEFF, INFIX, 1 variable.

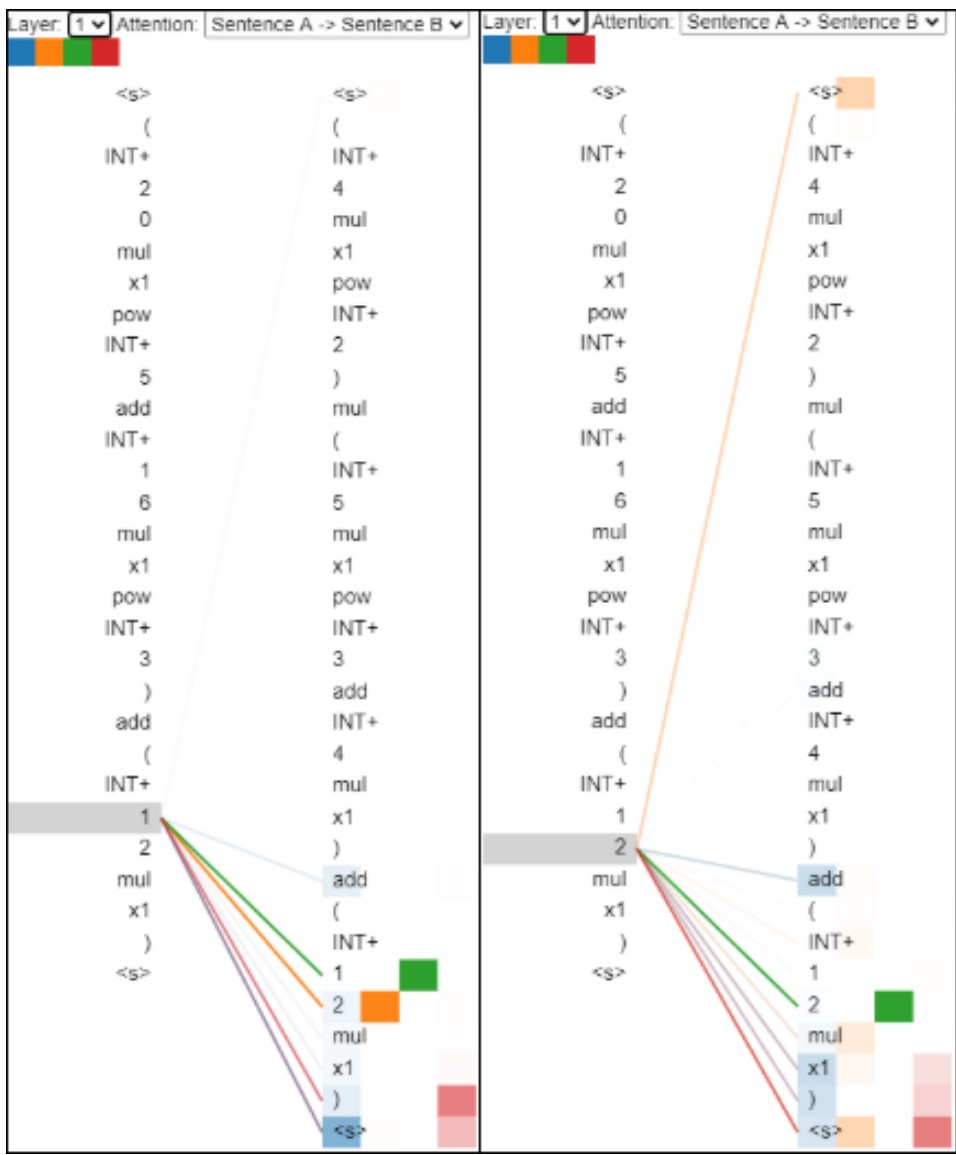

Figure 2: The layer 1 encoder-decoder attention for the coefficient 12 in the last product $(20 * x_1^5 + 16 * x_1 ** 3) + \underline{(12 * x_1)}$. It is expected, that in this step, this product remains unchanged and simply copied o the output. Therefore, we see that the layers learn to copy the coefficients directly by attending to the corresponding digits (i.e. 1 attends to 1 in the last product). Config: COARSE, SMALL COEFF, INFIX, 1 variable.

## B   ALGORITHMS

The polynomial sampling algorithms `buildProduct` and `buildFactor` are provided in Algorithms 1 and 2 respectively.

---

**Algorithm 2:** BuildFactor (Sampling A Factor)

---

**Input:** $x_{\mathcal{P}_i}, rdegree, rterms, rcoeff$
**Constraints:** `num_vars_fac, max_coeff_fac, max_terms_fac,`
        `max_degree_fac`
**Output:** A factor $f_j$, Number of terms $nterms_j$

1  Sample $nvar \in \{1, \ldots, \texttt{num\_vars\_fac}\}$
2  $cvars$ = Sample $nvar$ variables from $x_{\mathcal{P}_i}$        `// Variable set for this factor`
3  Sample $nterms \in \{1, \ldots, min(\texttt{max\_terms\_fac}, rterms)\}$
                                        `// # Terms for this factor`
4  Sample $\{d_k\}_{k=1}^{nterms}$, s.t. $d_k \in \{0, \ldots, min(\texttt{max\_degree\_fac}, rdegree)\}$
            `// Term degrees:  degree 0 allows for constant terms`
5  Sample $\{c_k\}_{k=1}^{nterms}$, s.t. $c_k \in \{1, \ldots, min(\texttt{max\_coeff\_fac}, rcoeff)\}$
                                        `// Term coefficients`
6  **for** $k \leftarrow 1$ **to** $nterms$ 1 **do**
7  $\quad$ selects $d[k]$ variables from $cvars$ with replacement
      `// E.g.  if` $d[k] = 4, cvars = [x_1, x_2]$. `May sample` $[x_1, x_2, x_1, x_1]$
8  $\quad$ Convert the selected $d[k]$ variables to a term `//` $t_k = c_k * x_1^3 * x_2$,
9  **end**
10  $f_j = \sum_{k=1}^{nterms} t_k$
11  **return** $f_j$;

---

## C   TABLE OF CONSTRAINTS AND NOTATIONS

We provide the full list of constraints and notations in Table 4.

| | | |
|---|---|---|
| Term Constraints | #Products | $nprod \in \{2, \ldots, \text{maxP}_{\mathcal{P}}\}$ |
| | #Factors in $P_i$ | $nfac_i \in \{2, \ldots, \text{maxf}_{\text{P}}\}, \forall i \in \{1, \ldots, nprod\}$ |
| | #Terms in $f_{ij}$ | $|\text{terms}(f_i)| \in \{1, \ldots, \text{maxT}_{\text{f}}\}, \forall f_i j \in f_{\mathcal{P}}$ |
| | #Terms in $\hat{P}_i$ | $|\text{terms}(\hat{P}_i)| \leq \text{maxT}_{\text{P}} \forall P_i \in \mathcal{P}_0$ |
| Degree Constraints | #Degree in $\hat{\mathcal{P}}$ | $\sum d_{mn} \leq \text{D}_{\mathcal{P}}, \forall m \, \hat{t_m} \in \text{terms}(\hat{P}), \forall n \, x_n \in \text{vars}(\hat{t_m})$ |
| | #Degree in $f_{ij}$ | $\sum d_{kl} \leq \text{D}_{\text{f}}, \forall k \, \text{terms}(f_{ij}), \forall f_{ij} \in f_{\mathcal{P}}$ |
| Variable Constraints | #Variables in $\mathcal{P}_0$ | $|x_{\mathcal{P}}| \leq \text{V}_{\mathcal{P}}$ |
| | #Variables in $P_i$ | $|\text{vars}(\text{P}_i)| \leq \text{V}_{\text{P}}, \forall P_i \in \mathcal{P}_0$ |
| | #Variables in $f_i$ | $|\text{vars}(\text{f}_{ij})| \leq \text{V}_{\text{f}}, \forall f_j \in f_{\mathcal{P}}$ |
| Coefficient Constraints | Coeff in $\hat{\mathcal{P}}$ | $\hat{a}_j \leq \text{C}_{\mathcal{P}}, \forall \hat{a}_j \in \text{coeffs}(\hat{P})$ |
| | Coeff in $\hat{P}_i$ | $\hat{a_{ij}} \leq \text{C}_{\text{P}}, \forall a \, \text{coeffs}(\hat{P}_i), \forall P_i \in \mathcal{P}_0$ |
| | Coeff in $f_i$ | $a_k \leq \text{C}_{\text{f}}, \forall a \, \text{coeffs}(f_{ij}), \forall f_{ij} \in f_{\mathcal{P}}$ |

Table 4: List of notations, and corresponding constraints that a generated polynomial satisfies.

## D   PROBLEM SPACE SIZE ESTIMATION

We present the problem space size estimates here in Table 5.

| Config | NVAR = 1 | | NVAR = 2 | |
|---|---|---|---|---|
| | Equation Size Estimate | Endpoint Size Estimate | Equation Size Estimate | Endpoint Size Estimate |
| **SMALL COEFF** | 104M | 8.24M | 184M | 27.4M |
| **MEDIUM COEFF** | 179M | 16.3M | 325M | 42.4M |
| **LARGE COEFF** | 289M | 32M | 507M | 68.8M |
| **NO BACKTRACK** | 324M | 54.9M | 538M | 104M |
| **MEDIUM DEG** | 459M | 67.4M | 902M | 144M |
| **MEDIUM TERMS** | 866M | 31.5M | 1.73B | 801M |

Table 5: Size Estimates for the problem space, after generating sets of size 5M.

# E  INPUT REPRESENTATION (ADDITIONAL RESULTS)

We present the results for FINE configuration for 2 variable setting here in Table 6. The errors made by the models for 1 Variable and 2 Variable settings are presented in Tables 7 and 8 respectively.

| Config | Proof Type | Endpoint | | #Train | Full Proof | | Stepwise | | Calibration | | | |
|---|---|---|---|---|---|---|---|---|---|---|---|---|
| | | #EE | Endpoint Acc. | | Full Proof Acc. | Greedy Stepwise Acc. | Top-1 Acc. | Beam-5 Acc. | Sure Rate | P | R | F1 |
| **SMALL COEFF** | Infix/Fine | 4.3M | 94.7 | 4.6M | 88.1 | 97.19 | 90.7 | 92.2 | 83.47 | 100 | 92.02 | 0.96 |
| | Prefix/Fine | 4.5M | 93.93 | 5.4M | 90.3 | 97.83 | 94.63 | 96.2 | 87.9 | 99.96 | 92.85 | 0.96 |
| **MEDIUM COEFF** | Infix/Fine | 7M | 95.3 | 4.4M | 82.2 | 96.25 | 94.28 | 95.76 | 86.24 | 100 | 91.47 | 0.96 |
| | Prefix/Fine | 5.2M | 92.77 | 2.9M | 72.4 | 93.6 | 91.53 | 94.33 | 81.97 | 100 | 89.55 | 0.94 |
| **LARGE COEFF** | Infix/Fine | 9M | 91.8 | 3.2M | 73 | 93.85 | 77.94 | 82.2 | 63 | 99.9 | 80.75 | 0.89 |
| | Prefix/Fine | 6.1M | 86.6 | 4.7M | 78.6 | 95.6 | 91.93 | 93.47 | 83.87 | 100 | 91.23 | 0.95 |
| **NO BACKTRACK** | Infix/Fine | 8.6M | 83.8 | 5.8M | 72.5 | 94.64 | 81.54 | 84.82 | 72.34 | 100 | 88.72 | 0.94 |
| | Prefix/Fine | 7.1M | 79.2 | 4.1M | 60.7 | 90.48 | 81.73 | 85.67 | 70.2 | 99.91 | 85.81 | 0.92 |
| **MEDIUM DEG** | Infix/Fine | 4.9M | 87.9 | 3.6M | 73.5 | 94.21 | 89.78 | 92.46 | 77.22 | 100 | 86.01 | 0.92 |
| | Prefix/Fine | 5.2M | 83.73 | 4.6M | 73.6 | 94.57 | 86.5 | 89.4 | 76.93 | 100 | 88.94 | 0.94 |
| **MEDIUM TERMS** | Infix/Fine | 8.5M | 90 | 4.8M | 64 | 92.98 | 79.04 | 81.86 | 66.92 | 99.88 | 84.56 | 0.92 |
| | Prefix/Fine | 6.6M | 87.07 | 4.5M | 62.9 | 92.74 | 86.4 | 89.07 | 73.67 | 100 | 85.26 | 0.92 |

Table 6: Results for FINE configuration for 2 Variables for Infix and Prefix representation (No curriculum, No annotation).

| Config | Proof Type | Full Proof | | Error Percentage | | | | | |
|---|---|---|---|---|---|---|---|---|---|
| | | Full Proof Accuracy | Greedy Stepwise Accuracy | First FacStep | Total FacStep | First MulStep | Total MulStep | First SumStep | Total SumStep |
| **SMALL COEFF** | Coarse/Infix | 95 | 98.83 | 8 | 9.43 | 88 | 84.91 | 4 | 5.66 |
| | Fine/Infix | **98.9** | **99.79** | 0 | 0 | 100 | 100 | 0 | 0 |
| | Coarse/Prefix | 95.3 | 98.97 | 4.26 | 4.08 | 72.34 | 71.43 | 23.4 | 24.49 |
| | Fine/Prefix | 96.9 | 99.4 | 9.68 | 9.68 | 77.42 | 77.42 | 12.9 | 12.9 |
| **MEDIUM COEFF** | Coarse/Infix | 92.8 | 98.24 | 1.39 | 1.25 | 95.83 | 92.5 | 2.78 | 6.25 |
| | Fine/Infix | 90.3 | 97.99 | 11.34 | 11.32 | 85.57 | 84.91 | 3.09 | 3.77 |
| | Coarse/Prefix | **93.6** | **98.58** | 3.12 | 2.94 | 95.31 | 95.59 | 1.56 | 1.47 |
| | Fine/Prefix | 91.7 | 98.37 | 2.41 | 2.33 | 96.39 | 96.51 | 1.2 | 1.16 |
| **LARGE COEFF** | Coarse/Infix | 82.1 | 95.97 | 3.35 | 3.02 | 93.85 | 91.46 | 2.79 | 5.53 |
| | Fine/Infix | 82.5 | 96.44 | 2.86 | 2.56 | 93.71 | 90.77 | 3.43 | 6.67 |
| | Coarse/Prefix | **83.5** | **96.25** | 4.24 | 3.78 | 93.94 | 92.97 | 1.82 | 3.24 |
| | Fine/Prefix | 82 | 96.32 | 3.33 | 2.97 | 90.56 | 86.63 | 6.11 | 10.4 |
| **NO BACKTRACK** | Coarse/Infix | 75.6 | 94.62 | 2.87 | 3.13 | 93.44 | 86.83 | 3.69 | 10.03 |
| | Fine/Infix | 74.5 | 94.76 | 3.14 | 3.56 | 93.33 | 78.63 | 3.53 | 17.81 |
| | Coarse/Prefix | **79.7** | **95.38** | 7.39 | 6.57 | 89.16 | 83.94 | 3.45 | 9.49 |
| | Fine/Prefix | 74.7 | 95.23 | 2.37 | 2.41 | 96.44 | 89.16 | 1.19 | 8.43 |
| **MEDIUM DEG** | Coarse/Infix | **92.8** | **98.26** | 5.56 | 6.02 | 86.11 | 79.52 | 8.33 | 14.46 |
| | Fine/Infix | 83.4 | 96.12 | 6.63 | 5.56 | 89.76 | 83.33 | 3.61 | 11.11 |
| | Coarse/Prefix | 87.7 | 96.82 | 4.07 | 3.57 | 93.5 | 90.71 | 2.44 | 5.71 |
| | Fine/Prefix | 90.6 | 97.92 | 8.51 | 7.55 | 89.36 | 87.74 | 2.13 | 4.72 |
| **MEDIUM TERMS** | Coarse/Infix | 72.7 | 93.99 | 25.64 | 24.18 | 73.26 | 69.72 | 1.1 | 6.1 |
| | Fine/Infix | 75.1 | 95.42 | 21.29 | 20.51 | 75.1 | 69.94 | 3.61 | 9.55 |
| | Coarse/Prefix | **76.3** | **95.78** | 7.59 | 8.71 | 88.61 | 84.67 | 3.8 | 6.62 |
| | Fine/Prefix | 74.8 | 95.55 | 14.68 | 16.76 | 79.76 | 74.28 | 5.56 | 8.96 |

Table 7: Errors for 1 variable in the COARSE and FINE configuration for both Infix and Prefix input representation. (No curriculum, No annotation).

| Config | Proof Type | Full Proof | | Error Percentage | | | | | |
|---|---|---|---|---|---|---|---|---|---|
| | | Full Proof Accuracy | Greedy Stepwise Accuracy | First FacStep | Total FacStep | First MulStep | Total MulStep | First SumStep | Total SumStep |
| SMALL COEFF | Coarse/Infix | 87.9 | 97.01 | 5.79 | 4.9 | 88.43 | 79.72 | 5.79 | 15.38 |
| | Fine/Infix | **88.1** | **97.19** | 8.4 | 7.98 | 75.63 | 68.1 | 15.97 | 23.93 |
| | Coarse/Prefix | **91.2** | **98.08** | 1.14 | 1.03 | 88.64 | 84.54 | 10.23 | 14.43 |
| | Fine/Prefix | 90.3 | 97.83 | 8.25 | 6.35 | 80.41 | 73.02 | 11.34 | 20.63 |
| MEDIUM COEFF | Coarse/Infix | 88.5 | 97.35 | 4.35 | 3.73 | 83.48 | 76.87 | 12.17 | 19.4 |
| | Fine/Infix | 82.2 | 96.25 | 2.25 | 1.83 | 76.4 | 68.81 | 21.35 | 29.36 |
| | Coarse/Prefix | **84.5** | **96.03** | 3.87 | 3.68 | 88.39 | 81.58 | 7.74 | 14.74 |
| | Fine/Prefix | 72.4 | 93.6 | 12.68 | 9.95 | 76.09 | 67.74 | 11.23 | 22.31 |
| LARGE COEFF | Coarse/Infix | 80.4 | 95.18 | 6.12 | 4.84 | 82.65 | 75.81 | 11.22 | 19.35 |
| | Fine/Infix | 73 | 93.85 | 11.85 | 8.74 | 70.74 | 62.3 | 17.41 | 28.96 |
| | Coarse/Prefix | **83.7** | **96.23** | 4.29 | 3.61 | 87.12 | 82.99 | 8.59 | 13.4 |
| | Fine/Prefix | 78.6 | 95.6 | 5.14 | 4.2 | 81.31 | 74.43 | 13.55 | 21.37 |
| NO BACKTRACK | Coarse/Infix | **72.7** | **93.13** | 4.4 | 3.15 | 87.55 | 75.79 | 8.06 | 21.07 |
| | Fine/Infix | 72.5 | 94.64 | 3.27 | 2.54 | 85.09 | 73.79 | 11.64 | 23.66 |
| | Coarse/Prefix | **63.2** | **89.87** | 3.26 | 2.24 | 91.3 | 78.73 | 5.43 | 19.03 |
| | Fine/Prefix | 60.7 | 90.48 | 2.29 | 1.58 | 89.31 | 72.64 | 8.4 | 25.79 |
| MEDIUM DEG | Coarse/Infix | **80.5** | **95.13** | 6.67 | 5.44 | 81.54 | 71.97 | 11.79 | 22.59 |
| | Fine/Infix | 73.5 | 94.21 | 7.17 | 6.55 | 68.3 | 57.83 | 24.53 | 35.61 |
| | Coarse/Prefix | **83.4** | **96.41** | 4.82 | 4.19 | 81.33 | 75.39 | 13.86 | 20.42 |
| | Fine/Prefix | 73.6 | 94.57 | 7.58 | 6.38 | 75.38 | 67.48 | 17.05 | 26.14 |
| MEDIUM TERMS | Coarse/Infix | 64 | 92.03 | 25 | 19.05 | 72.5 | 66.5 | 2.5 | 14.45 |
| | Fine/Infix | 64 | 92.98 | 13.61 | 8.62 | 79.44 | 69.59 | 6.94 | 21.79 |
| | Coarse/Prefix | **67.8** | **93.58** | 10.25 | 7.69 | 87.89 | 80.98 | 1.86 | 11.32 |
| | Fine/Prefix | 62.9 | 92.74 | 9.16 | 5.97 | 85.71 | 74.37 | 5.12 | 19.65 |

Table 8: Errors for 2 variables in the COARSE and FINE configuration for both Infix and Prefix input representation. (No curriculum, No annotation).

## F  ANNOTATED PROOF (ADDITIONAL RESULTS)

We present the results for COARSE and FINE configuration for 2 variable setting for annotated proofs here in Table 9. The errors made by the models for 1 Variable and 2 Variable settings are presented in Tables 10 and 11 respectively.

| Config | Proof Type | Endpoint | | # Train Examples | Full Proof | | Stepwise | | Calibration | | | |
|---|---|---|---|---|---|---|---|---|---|---|---|---|
| | | # Endpoint Examples | Endpoint Accuracy | | Full Proof Accuracy | Greedy Stepwise Accuracy | Top-1 Accuracy | Beam-5 Accuracy | Sure Rate | P | R | F1 |
| SMALL COEFF | Fine | 4.3M | 94.7 | 3.6M | 82.3 | 97.93 | 86.47 | 87.5 | 81.83 | 100 | 94.64 | 0.97 |
| | Coarse | | | 5.1M | **85** | **98.31** | 93.5 | 94.03 | 90.27 | 100 | 96.54 | 0.98 |
| MEDIUM COEFF | Fine | 7M | 95.3 | 5.4M | 78.8 | 97.78 | 93.8 | 94.5 | 90.2 | 99.93 | 96.09 | 0.98 |
| | Coarse | | | 5M | **80.1** | **97.69** | 89.37 | 90.27 | 86.77 | 99.96 | 97.05 | 0.98 |
| LARGE COEFF | Fine | 9M | 91.8 | 4.1M | 70.1 | 96.59 | 84.8 | 86.63 | 77.77 | 99.83 | 91.55 | 0.96 |
| | Coarse | | | 4M | **73.2** | **93.8** | 92.77 | 93.8 | 87.23 | 100 | 94.04 | 0.97 |
| NO BACKTRACK | Fine | 8.6M | 83.8 | 3.5M | **46.5** | **92.93** | 84.9 | 87.67 | 74.5 | 99.96 | 87.71 | 0.93 |
| | Coarse | | | 6.7M | **65.5** | **95.7** | 67.8 | 69.37 | 63.3 | 99.79 | 93.17 | 0.96 |
| MEDIUM DEG | Fine | 4.9M | 87.9 | 3.9M | 59.6 | 95.28 | 94.13 | 95.7 | 86.4 | 100 | 91.78 | 0.96 |
| | Coarse | | | 4.1M | **65.1** | **95.61** | 85.43 | 87.43 | 78.2 | 99.96 | 91.49 | 0.96 |
| MEDIUM TERMS | Fine | 8.5M | 90 | 4.8M | **56.9** | **95.7** | 92.4 | 93.83 | 85.77 | 99.88 | 92.71 | 0.96 |
| | Coarse | | | 4.2M | 52.8 | 94.57 | 84 | 85.93 | 75.93 | 99.82 | 90.24 | 0.95 |

Table 9: Results for FINE and COARSE configurations for 2 Variables for annotated proofs (No curriculum).

## G  FULLY SYMBOLIC PROOFS

As > 80% of the errors occurred in multiplication step, we separately tested the Transformer's ability to do arithmetic, by creating datasets involving multiplication and addition of 4-digit and 9-digit numbers. While the models quickly achieved an accuracy of close to 99% for addition; for multiplication, they could not go beyond even 1% after seeing 2M examples. Hence, we envision a setting where polynomial simplification steps only involve symbolic addition and multiplication, without any arithmetic manipulation. For example, instead of multiplying 3 and 4 as 12, the model will output $c_1 * c_2$ given coefficients $c_1$ and $c_2$. The results for 1 Variable setting are presented in Table 12. Here, MEDIUM COEFF and MEDIUM DEGREE denote the same configuration as the case with integer coefficients. The only difference being that the limits of coefficients no longer apply. The

| Config | Proof Type | Full Proof | | Error Percentage | | | | | | | |
| | | Full Proof Accuracy | Greedy Stepwise Accuracy | First FacStep | Total FacStep | First MulStep | Total MulStep | First SumStep | Total SumStep | First MarkStep | Total MarkStep |
|---|---|---|---|---|---|---|---|---|---|---|---|
| SMALL COEFF | Fine | 88.5 | 98.82 | 3.48 | 2.99 | 89.57 | 83.58 | 6.09 | 11.19 | 0.87 | 2.24 |
| | Coarse | 91.9 | 99.16 | 1.23 | 1.19 | 98.77 | 96.43 | 0 | 1.19 | 0 | 1.19 |
| MEDIUM COEFF | Fine | 78.6 | 97.66 | 18.69 | 15.19 | 74.77 | 74.44 | 3.27 | 6.67 | 3.27 | 3.7 |
| | Coarse | 84.2 | 98.29 | 4.43 | 4.65 | 84.81 | 84.88 | 6.33 | 5.81 | 4.43 | 4.65 |
| LARGE COEFF | Fine | 75.5 | 97.37 | 11.43 | 9.21 | 72.65 | 66.35 | 10.61 | 19.68 | 5.31 | 4.76 |
| | Coarse | 80.3 | 97.86 | 5.58 | 5.86 | 90.86 | 87.39 | 1.02 | 4.5 | 2.54 | 2.25 |
| NO BACKTRACK | Fine | 68 | 96.78 | 7.19 | 6.46 | 86.56 | 78.54 | 5.62 | 12.71 | 0.62 | 2.29 |
| | Coarse | 59.7 | 95 | 6.2 | 5.25 | 88.09 | 76.88 | 3.72 | 15.41 | 1.99 | 2.45 |
| MEDIUM DEG | Fine | 76 | 97.37 | 11.67 | 10.85 | 82.5 | 80.34 | 3.75 | 6.78 | 2.08 | 2.03 |
| | Coarse | 78.7 | 97.38 | 6.1 | 5.84 | 86.38 | 81.32 | 4.69 | 9.73 | 2.82 | 3.11 |
| MEDIUM TERMS | Fine | 70.4 | 97.48 | 16.89 | 16.27 | 75 | 69.14 | 3.72 | 8.85 | 4.39 | 5.74 |
| | Coarse | 66.2 | 96.34 | 25.44 | 25.28 | 68.05 | 63.48 | 2.37 | 5.81 | 4.14 | 5.43 |

Table 10: Errors for FINE and COARSE configurations for 1 Variable for annotated proofs (No curriculum).

| Config | Proof Type | Full Proof | | Error Percentage | | | | | | | |
| | | Full Proof Accuracy | Greedy Stepwise Accuracy | First FacStep | Total FacStep | First MulStep | Total MulStep | First SumStep | Total SumStep | First MarkStep | Total MarkStep |
|---|---|---|---|---|---|---|---|---|---|---|---|
| SMALL COEFF | Fine | 82.3 | 97.93 | 4.52 | 3.07 | 86.44 | 68.97 | 7.34 | 24.14 | 1.69 | 3.83 |
| | Coarse | 85 | 98.31 | 2 | 1.68 | 88.67 | 78.21 | 8 | 18.44 | 1.33 | 1.68 |
| MEDIUM COEFF | Fine | 78.8 | 97.78 | 8.96 | 6.79 | 80.19 | 68.21 | 9.43 | 22.5 | 1.42 | 2.5 |
| | Coarse | 80.1 | 97.69 | 9.05 | 7.79 | 87.94 | 80.33 | 3.02 | 10.66 | 0 | 1.23 |
| LARGE COEFF | Fine | 70.1 | 96.59 | 13.38 | 10 | 69.9 | 59.32 | 13.38 | 25.45 | 3.34 | 5.23 |
| | Coarse | 73.2 | 96.66 | 10.45 | 7.84 | 79.85 | 70.87 | 7.84 | 18.49 | 1.87 | 2.8 |
| NO BACKTRACK | Fine | 46.5 | 92.93 | 9.16 | 5.15 | 74.21 | 57.9 | 14.58 | 33.69 | 2.06 | 3.25 |
| | Coarse | 65.5 | 95.7 | 3.19 | 2.61 | 90.14 | 77.31 | 5.22 | 18.27 | 1.45 | 1.81 |
| MEDIUM DEG | Fine | 59.6 | 95.28 | 7.43 | 5.48 | 72.03 | 57.1 | 15.35 | 32.9 | 5.2 | 4.52 |
| | Coarse | 65.1 | 95.61 | 6.88 | 5.26 | 78.51 | 67.79 | 11.46 | 24.63 | 3.15 | 2.32 |
| MEDIUM TERMS | Fine | 56.9 | 95.7 | 21.58 | 13.3 | 67.29 | 57.72 | 8.58 | 23.96 | 2.55 | 5.02 |
| | Coarse | 52.8 | 94.57 | 23.94 | 15.6 | 68.22 | 62.77 | 3.6 | 16.67 | 4.24 | 4.96 |

Table 11: Errors for FINE and COARSE configurations for 2 Variable for annotated proofs (No curriculum).

errors made by the model for each kind of step are summarized in Table 13.

We observe that the proof accuracy is about 20% less than the non-symbolic models. This could be because the intermediate polynomials in the simplification sequence become very long with symbolic coefficients.

| Config | Proof Type | Endpoint | | #Train | Full Proof | |
| | | #EE | Endpoint Acc. | | Full Proof Acc. | Greedy Stepwise Acc. |
|---|---|---|---|---|---|---|
| **MEDIUM COEFF** | Coarse/Infix | 5M | 93.5 | 4.3M | 78.5 | 94.19 |
| | Fine/Infix | | | 2.8M | 63.5 | 90.64 |
| | Coarse/Prefix | 4.9M | 89.77 | 3.7M | 70.9 | 91.53 |
| | Fine/Prefix | | | 4.3M | 70.9 | 93.2 |
| **MEDIUM DEGREE** | Coarse/Infix | 5.6M | 88 | 3.7M | 65.2 | 89.55 |
| | Fine/Infix | | | 6.3M | 75.5 | 94.59 |
| | Coarse/Prefix | 6.3M | 83.93 | 3.4M | 57.6 | 85.98 |
| | Fine/Prefix | | | 6.7M | 67.7 | 92.76 |

Table 12: Results for 1 Variable Symbolic Coeff setting. (No curriculum, No annotation).

# H   OUT-OF-DISTRIBUTION EVALUATION

We present the results for Out-of-Distribution evaluation here. Table 14 contains results for best 2 variable models (Prefix/Coarse) tested on 1 Variable setting.

Table 15 contains results for best 1 variable models (Prefix/Coarse) tested on SMALL, MEDIUM and LARGE coefficient setting. As expected, the SMALL and MEDIUM models perform much worse when tested on higher coefficients.

We also evaluated the best 1 variable models (Prefix/Coarse) on MEDIUM DEGREE and TERMS set-

| Config | Proof Type | Full Proof | | Error Percentage | | | | | |
|---|---|---|---|---|---|---|---|---|---|
| | | Full Proof Accuracy | Greedy Stepwise Accuracy | First FacStep | Total FacStep | First MulStep | Total MulStep | First SumStep | Total SumStep |
| MEDIUM COEFF | Coarse/Infix | 78.5 | 94.19 | 3.72 | 2.82 | 77.67 | 67.25 | 18.6 | 29.93 |
| | Fine/Infix | 63.5 | 90.64 | 3.29 | 2.29 | 79.73 | 67.43 | 16.99 | 30.28 |
| | Coarse/Prefix | 70.9 | 91.53 | 2.06 | 1.45 | 76.29 | 64.01 | 21.65 | 34.54 |
| | Fine/Prefix | 70.9 | 93.2 | 1.37 | 0.97 | 73.88 | 66.1 | 24.74 | 32.93 |
| MEDIUM DEGREE | Coarse/Infix | 65.2 | 89.55 | 2.01 | 1.41 | 85.06 | 71.49 | 12.93 | 27.11 |
| | Fine/Infix | 75.5 | 94.59 | 0.41 | 0.31 | 86.12 | 77.02 | 13.47 | 22.67 |
| | Coarse/Prefix | 57.6 | 85.98 | 6.6 | 4.34 | 83.25 | 65.42 | 10.14 | 30.24 |
| | Fine/Prefix | 67.7 | 92.76 | 1.86 | 1.39 | 88.54 | 79.35 | 9.6 | 19.26 |

Table 13: Errors made by models in 1 Variable Symbolic Coeff setting. (No curriculum, No annotation).

tings, to check generalization with respect to # terms and degree of polynomial. Table 16 contains results for the same. The MEDIUM COEFF model is not able to generalize to more terms or polynomials of higher degree.

| Config | Train/Test= 2 Var/1 Var | | Train/Test= 1 Var/1 Var | | Train/Test= 2 Var/2 Var | |
|---|---|---|---|---|---|---|
| | Full Proof Acc. | Greedy Stepwise Acc. | Full Proof Acc. | Greedy Stepwise Acc. | Full Proof Acc. | Greedy Stepwise Acc. |
| SMALL COEFF | 95.34 | 99.12 | 95.3 | 98.97 | 91.2 | 98.08 |
| MEDIUM COEFF | 87.4 | 97.11 | 93.6 | 98.58 | 84.5 | 96.03 |
| LARGE COEFF | 89.4 | 97.13 | 83.5 | 96.25 | 83.7 | 96.23 |
| NO BACK TRACK | 84.2 | 98.29 | 79.7 | 95.38 | 63.2 | 89.87 |
| MEDIUM DEG | 87.7 | 97.83 | 87.7 | 96.82 | 83.4 | 96.41 |
| MEDIUM TERMS | 78.5 | 96.16 | 76.3 | 95.78 | 67.8 | 93.58 |

Table 14: Results for OOD Testing. NVAR = 2 COARSE/PREFIX models tested on corresponding NVAR = 1 setting (No curriculum, No annotation).

| Train Config | Test Config | | | | | |
|---|---|---|---|---|---|---|
| | SMALL COEFF | | MEDIUM COEFF | | LARGE COEFF | |
| | Full Proof Acc. | Greedy Stepwise Acc. | Full Proof Acc. | Greedy Stepwise Acc. | Full Proof Acc. | Greedy Stepwise Acc. |
| SMALL COEFF | 95.3 | 98.97 | 33.4 | 69.05 | 31 | 68.02 |
| MEDIUM COEFF | 96.6 | 99.29 | 93.6 | 98.58 | 33.6 | 68.96 |
| LARGE COEFF | 95.8 | 99.1 | 94.4 | 98.64 | 83.5 | 96.25 |

Table 15: OOD Testing: Prefix/Coarse 1 Variable Models tested on various coefficient limit configurations (SMALL, MEDIUM and COARSE). (No curriculum, No annotation).

# I CURRICULUM LEARNING

Learning the simplification steps should entail learning the sub-tasks, such as addition and multiplication (of numeric coefficients and symbolic variables); where multiplying variables precludes learning to add exponents of similar variables. As these sub-tasks are well-defined and dependencies among them are clear, we explore different types of curriculums based on the Mastering-Rate-based (MR) curriculum learning algorithm proposed in Willems et al. (2020). Authors in Willems et al.

| Train Config | Test Config | | | | | |
|---|---|---|---|---|---|---|
| | MEDIUM COEFF | | MEDIUM DEG | | MEDIUM TERMS | |
| | Full Proof Acc. | Greedy Stepwise Acc. | Full Proof Acc. | Greedy Stepwise Acc. | Full Proof Acc. | Greedy Stepwise Acc. |
| MEDIUM COEFF | 93.6 | 98.58 | 20.8 | 47.77 | 26.1 | 54.65 |
| MEDIUM DEG | 94.8 | 98.93 | 87.7 | 96.82 | 25.5 | 54.39 |
| MEDIUM TERMS | 92.7 | 96.87 | 18.6 | 46.97 | 76.3 | 95.78 |

Table 16: Prefix/Coarse 1 Variable Models tested on various #term and degree configurations (MEDIUM DEGREE and MEDIUM TERMS). (No curriculum, No annotation).

(2020) define curriculum learning by 1) a *curriculum* i.e. a set of tasks $\mathcal{C} = \{c_1, \ldots, c_n\}$, where a task is set of examples of similar type with a sampling distribution, and 2) a *program* which for each training step defines the tasks to train the learner given its learning state and the curriculum. Formally, the program $d : \mathbb{N} \to \mathcal{D}^{\mathcal{C}}$, is a sequence of distributions over $\mathcal{C}$. The authors estimate the *program* function through an *attention* function which defines attention over the tasks at a time-step, and an *attention-to-distribution converter* which converts the attention to a distribution over $\mathcal{C}$. Authors observe that other algorithms (Matiisen et al., 2019; Graves et al., 2017) are special cases of the above setting with different choices for *program*.

To learn on tasks that are *learnable but not learnt yet*, authors define an *ordered curriculum* $\mathcal{O}^{\mathcal{C}}$ which is a directed graph over tasks in $\mathcal{C}$. An edge from A to B indicates that learning task A before B is preferable. For supervised learners, the learnability for each task depends on mastering rate ($\mathcal{M}_c(t)$) computed from the normalized mean accuracy for that task at time-step $t$. At each time-step, the MR algorithm computes attention over a task ($a_c(t)$) from mastering rates of its ancestors and successors. During training to sample batches, a hyperparameter $N_b$ for the curriculum determines the number of batches to be considered at a step, before re-computing the attention over tasks. Using the *program* $d$, we first sample $N_b * b$ examples from tasks in $\mathcal{C}$. The model is then trained on randomly sampled $N_b$ minibatches are sampled updating the mastering rates.

For polynomial simplification for 1 variable, we define the following tasks ADD, MUL2, MUL3, SCOEFF and MIXED. For ADD, only one factor per product is allowed, so there is no multiplication. For MUL2 and MUL3 only 1 product is allowed with maximum two factors and three factors respectively. SCOEFF points to the SMALL COEFF configuration and MIXED is the final variable size configuration of the target variable configuration. We define the following curriculums:
- C: {(ADD, MUL3), (MUL3, MIXED), (ADD, MIXED)}.
- C2: {(ADD, MUL2), (MUL2, MUL3), (MUL3, MIXED), (ADD, MIXED)}.
- C4: {(ADD, MUL2), (MUL2, MUL3), (MUL3, SCOEFF), (ADD, SCOEFF) (SCOEFF, MIXED)}.
For all our experiments, we use the MR algorithm with gAmax Linreg A2D converter functions described in Willems et al. (2020). Model parameters and the training configurations remains the same as before[6]. We show the results in Table 17 for COARSE configuration. As coefficient size grows from SMALL, MEDIUM, LARGE to NO BACKTRACK - the improvements in full proof accuracy steadily increase from $1\%$ to $10.84\%$. For NO BACKTRACK, the improvement in top-1 accuracy is by $20\%$ from a no curriculum setting. However, we observe for MEDIUM TERMS, there is a drop in accuracy for all curriculums and input representations. It is possible that, more carefully designed curriculums may improve the results. There is no conceivable pattern observed for infix or prefix representations. However, compared to learning without curriculum, the improvement observed for infix representation is larger than prefix.

---

[6]We use $N_b$ as 10. For other default parameters in CL, please check `github.com/lcswillems/automatic-curriculum`.

| | | | | Full Proof | | Step-wise | | Calibration | | | |
|---|---|---|---|---|---|---|---|---|---|---|---|
| | | Curri culum | #Train | Full Proof Accuracy | Stepwise Accuracy | Top-1 Acc | Beam-5 Acc | Sure Rate | P | R | F1 |
| Small Coeff | Infix | C | 2.8M | 94.38 | 98.76 | 94.84 | 96.68 | 89.36 | 100 | 94.22 | 0.97 |
| | | C2 | 2M | **95.98** | **99.0** | 91.64 | 93.24 | 86.16 | 99.9 | 93.98 | 0.97 |
| | Prefix | C | 2.02M | 94.26 | 98.65 | 77.76 | 80.46 | 70.62 | 99.94 | 90.77 | 0.95 |
| | | C2 | 2.29M | 94.6 | 98.56 | 93.44 | 95.28 | 88.02 | 99.89 | 94.09 | 0.97 |
| Medium Coeff | Infix | C2 | 3.9M | **95.44** | **99.02** | 94.86 | 96.44 | 91.18 | 100 | 96.12 | 0.98 |
| | | C4 | 2M | 93.86 | 98.59 | 88.22 | 90.24 | 84.68 | 99.91 | 95.90 | 0.98 |
| | Prefix | C2 | 3.7M | 94.78 | 98.82 | 91.98 | 93.66 | 88.08 | 99.93 | 95.69 | 0.98 |
| | | C4 | 4.4M | 94.8 | 98.87 | 85.3 | 87.82 | 80.62 | 99.98 | 94.49 | 0.97 |
| Large Coeff | Infix | C2 | 6.9M | 91.26 | 97.92 | 96.4 | 98.06 | 90.24 | 99.89 | 93.51 | 0.97 |
| | | C4 | 7.6M | 91.62 | 98.16 | 91.54 | 93.3 | 87.38 | 99.84 | 95.3 | 0.98 |
| | Prefix | C2 | 6.5M | 92.2 | 98.31 | 85.38 | 87.78 | 81.42 | 99.95 | 95.32 | 0.98 |
| | | C4 | 6.97M | **92.46** | **98.42** | 91.3 | 93.34 | 87.54 | 100.0 | 95.88 | 0.98 |
| No Backtrack | Infix | C2 | 4.8M | 86.44 | 97.27 | 93.68 | 95.46 | 88.72 | 99.98 | 94.68 | 0.97 |
| | | C4 | 5.1M | 85.96 | 97.21 | 94.64 | 96.1 | 89.5 | 100 | 94.57 | 0.97 |
| | Prefix | C2 | 7M | 86.16 | 97.30 | 82.24 | 84.44 | 77.46 | 99.95 | 94.14 | 0.97 |
| | | C4 | 5.5M | **86.48** | **97.45** | 92.6 | 94.3 | 87.78 | 99.95 | 94.75 | 0.97 |
| Medium Degree | Infix | C2 | 3.5M | 87.12 | 97.01 | 84.16 | 87.44 | 78.46 | 99.95 | 93.18 | 0.96 |
| | | C4 | 3.4M | 94.12 | 98.65 | 90.62 | 81.984 | 86.66 | 99.93 | 95.56 | 0.98 |
| | Prefix | C2 | 5.35M | **94.28** | **98.71** | 80.8 | 82.84 | 75.76 | 100 | 93.51 | 0.97 |
| | | C4 | 3.5M | 92.38 | 98.30 | 83.7 | 85.48 | 78.94 | 99.92 | 94.24 | 0.97 |
| Medium Terms | Infix | C2 | 4.4M | 59.54 | 75.76 | 65.6 | 69.56 | 60.84 | 95.36 | 88.45 | 0.92 |
| | | C4 | 3.8M | 56.94 | 76.72 | 69.84 | 73.44 | 60.76 | 97.5 | 84.82 | 0.91 |
| | Prefix | C2 | 2.8M | 41.84 | 51.24 | 40.62 | 45.36 | 36.9 | 92.57 | 84.10 | 0.88 |
| | | C4 | 3.37M | 49.02 | 65.41 | 58.56 | 64.64 | 45.44 | 96.83 | 75.14 | 0.85 |

Table 17: Curriculum Learning results for 1 variable for the COARSE configuration for both Infix and prefix representations.

