# OpenReview forum: "Do Transformers Understand Polynomial Simplification? "
_ICLR.cc/2021/Conference — Reject_

### Official Review · AnonReviewer4 · 2020-10-17
**Interesting study of the behavior of Transformers on symbolic tasks**

**Rating:** 6
**Confidence:** 4

**Review:**

The paper "Do Transformers Understand Polynomial Simplification?" introduces a new reasoning task (convert polynomials into a normal form) and studies the performance and errors of Transformers on this task. The task itself is quite simple: given a randomly generated term involving small constants, variables, additions, and multiplications, bring the term into a well-defined normal form. Each task has to be solved in a unique sequence of steps. The authors study the performance of Transformers to either simplify the expressions step by step or to predict the simplified version directly.

Contributions I highly appreciate:
- Contrasting step-wise generation of proofs vs end-point prediction.
- Contrasting different representations of formulas: prefix notation appears to be better than infix notation.
- Observation that transformers struggle with multiplication.

Questions to the authors:
- The task is pretty simple compared to other (mathematical) reasoning tasks studied in the literature. The task is also not very useful as a prediction target, because it can be solved with a simple algorithm, as far as I can see. The paper essentially claims that this allows us to study Transformers better. What insights were produced here that couldn't have been produced on more challenging benchmark from the literature?
- Why is it important for your paper that proofs are unique? Do we simply train Transformers to execute a simple algorithm? If so, how does that relate to the existing theorem proving and reasoning approaches where we usually have many possible proofs for a task.
- What is the point of "establishing baselines" if the task is essentially solved already with moderately-sized Transformers?
- What can be learned from the curriculum learning experiments? The way the curriculum is defined here appears to essentially require a synthetic benchmark for which we can generate examples of different hardness levels. So how could this help for real problems?
- Introduction and Conclusion: The paper claims that training on the two variable case leads to better performance on the one variable case than training on it directly. This sounds very much like an artifact that could stem from the lack of training data or so. How is this possible?


Minor comments:

p.1, pargraph 1: I believe Hahn et al also propose an "end-to-end" task using Transformers instead of embedding their approach in an existing neuro-symbolic system.

p.1, paragraph 2: The reference to Lample & Charton is slightly off: It was published in ICLR 2020.

p.2, paragraph 1: "we observe that the system can understand candidate sub-expressions ..." I am always wary of the use of "understand" in the context of neural networks, as it is not very clear what it means.

p2. related work: I do not understand how you contrast your work to the existing theorem proving works: There are a number of neural theorem provers (HOList, GamePad, GPT-f, and probably some more) that also generate proofs step by step. They might employ more advanced search ideas, but I think it would be good to state why your paper does not want to go in this direction.

p.2, Section 3: How does Sympy "ensure correctness"? There could be bugs in the code even if you didn't write the code yourself?

p.2, footnote: "an unique" -> "a unique"

p.3, last paragraph: remove "Hence".

Additional details about the training setup would be appreciated. For example, how many training steps/epochs did you train for?

---

> ### Author Response · Authors · 2020-11-19
> **Response to AnonReviewer4**
>
> Thank you for your kind and encouraging comments. Please find our response for individual comments below. We put your comments in quotes, followed by our response.
>
> “The task is also not very useful as a prediction target, because it can be solved with a simple algorithm, as far as I can see. The paper essentially claims that this allows us to study Transformers better. What insights were produced here that couldn't have been produced on more challenging benchmark from the literature?” : We agree that the final goal of this line of research is to design models that can learn to solve problems that we don't already know how to solve. Several recent works in ATP frameworks cited in the paper have attempted to do this. The accuracies achieved remain low. This could be for various reasons including deficiencies of the models and/or of data. To understand the situation better, it seems reasonable to first check if our models are able to solve much simpler problems where we have fine-grained control over various aspects of the dataset (this is impossible for ATP frameworks). For some examples of insights from our work, please see our common response.
>
> Unique Proofs: In keeping with our general philosophy of having a very simple setting, unique proofs simplify the task of the Transformer as at each step it has a unique step to perform.
> (If the proofs are not unique, then one could use RL/MCTS-based approaches as is done in many papers or learn the most probable next steps as in language models.)
>
> “What is the point of "establishing baselines" if the task is essentially solved already with moderately-sized Transformers?”: It is true that the task can be solved by a simple algorithm, however the task using Transformers as the model is not solved (Low accuracies in Table 1 and 2). Thus, our results can serve as a baseline for this problem.
>
> “What can be learned from the curriculum learning experiments? The way the curriculum is defined here appears to essentially require a synthetic benchmark for which we can generate examples of different hardness levels. So how could this help for real problems?”
> This is a good point. Certainly new curriculums will be needed for real problems.
> CL Experiments: First a note that, by using synthetic dataset generation, our goal was to eliminate the concerns such as “lack of data”.  Since, even in this setting, Transformers perform poorly, in many configurations, we wanted to employ traditional techniques such as CL to see whether such available techniques may improve the performance of Transformers. We observe that for many configurations, there is surprising improvement when CL is employed carefully. But, its inability to show a steady performance improvement again shows why more work is required for improving Transformers’ performance even for a seemingly simpler task of polynomial simplification.
>
> “Introduction and Conclusion: The paper claims that training on the two variable case leads to better performance on the one variable case than training on it directly. This sounds very much like an artifact that could stem from the lack of training data or so. How is this possible?”:
> While we don’t have an explanation for this observation, we remark that:
> it can’t be an artefact of the data. Both 1-var and 2-var models converge after similar number of steps (#Train column in Table 1 and 2). Thus, 1-var models see more single variable polynomials than 2-var models. But the 2-var model gets polynomials sampled from the 2 var distribution. (Table 1 and 2) It could happen that with 2 variables, the task of collecting variables together is better exemplified. In the sense that the tougher task model is performing good on a simpler task
>
> “p.2, paragraph 1: "we observe that the system can understand candidate sub-expressions ..." I am always wary of the use of "understand" in the context of neural networks, as it is not very clear what it means.”: We agree. A better title would have been “Can Transformers Perform Polynomial Simplification.”
>
> “For example, how many training steps/epochs did you train for?”
> As the data is generated on the fly, number of examples seen = number of training steps. #EE and #Train columns in the result tables correspond to the number of training steps. We will update the draft if we see more details are required for reproduction of the results.

---

### Official Review · AnonReviewer2 · 2020-10-26
**Nice analysis with some more possibilities**

**Rating:** 6
**Confidence:** 4

**Review:**

The authors analyze the performance of Transformer models on simplifying polynomials and - importantly - generating proofs at the same time. This is a very nice idea that allows to study the performance of Transformers in depth and at the same time in an important setting where verification is performed as part of running the model. And the authors show a strong baseline, with models performing very well in a number of settings. A few areas seem to have been neglected though. For one, the authors only train a 4-layer 4-head model, which is quite small as far as Transformers go. Maybe it's irrelevant for this problem - but having at least one bigger model as a point of comparison would be good. Next, the out-of-distribution question warrants more experiments. Can the Transformers simplify polynomials with way more factors than trained on? With a higher number of variables? Higher degrees? The authors also show that one main problem for Transformers is learning to multiply the coefficients. But - assuming this reviewer understood correctly - the authors do not apply the proof requirement to multiplication. E.g., for "12*3" the model has to immediately output "36" rather than "10*3 + 2*3 = 30 + 6 = 36". Maybe this could help the Transformer learn and be more resilient to coefficient size? So while the current version of the paper is ok, there are a few areas for improvement which prevent it from being a clear accept.

---

> ### Author Response · Authors · 2020-11-19
> **Response to AnonReviewer2**
>
> Dear Reviewer, thank you for your encouraging comments. Please find our response below.
> “For one, the authors only train a 4-layer 4-head model, which is quite small as far as Transformers go.”: We plan to perform these experiments in future. However, as mentioned in the general response, such a choice does not conflict with our premise and the obtained insights.
>
> “Can the Transformers simplify polynomials with way more factors than trained on? With a higher number of variables? Higher degrees?”: For OOD evaluation, we evaluate by increasing more factors, higher coefficients, and higher exponents. The comparison of MEDIUM COEFF, MEDIUM TERMS & MEDIUM DEGREE (Table 16) shows this data. For 2 variables and other settings, please check Tables 14, 15.
>
> Finer Steps in Arithmetic Multiplication question: You are correct in stating that we do not add such finer steps for arithmetic multiplication. However, our results show (Table 9 for annotated proofs) that Transformers suffer on longer proofs. Since, the above modification will generate even longer proofs, we suspect the setting will lead to even worse results. Having said that, we do plan to perform experiments where the difficulty in integer factorization is “outsourced” by making an external program (a standard integer multiplication program) perform integer multiplication and letting the Transformers handle only symbolic manipulation. As the experimentation over all variations will take time, we believe we will be able to include the results in the final version upon acceptance.

---

> ### Comment · AnonReviewer2 · 2020-11-20
> **Thank you for your response**
>
> Thank you for pointing out Table 16 - it was (and still partially is) a bit hard to understand all the OOD settings, it could be worth re-thinking the division into small/medium, but understandably there needs to be some classification. It is also not clear whether the small drop from coarse to fine in Table 9 would generalize to factoring out multiplication, as it would possibly significantly simplify multiplication (in contrast to the coarse->fine change that is a much less algorithmic simplification). It is also possible that a larger model would improve on handling long sequences, it would certainly be worth trying - even if indeed it shouldn't affect the main points.

---

> > ### Author Response · Authors · 2020-11-23
> > **Thank you for your Response**
> >
> > "It is also possible that a larger model would improve on handling long sequences, it would certainly be worth trying - even if indeed it shouldn't affect the main points."
> >
> > We agree with your suggestion, and will experiment with it.

---

### Official Review · AnonReviewer3 · 2020-10-28
**Many experiments but not sufficiently interesting new insights**

**Rating:** 4
**Confidence:** 4

**Review:**

The paper studies the capability of the transformer architecture to
perform rewriting to normal form in a simplified polynomial setting.

It is a continuation of the research by Piotrowski et al (PUBK) in the
area of using neural nets to do symbolic rewriting, followed later by
Lample&Charton (LC).

Several datasets are generated, using various
constraints on the sizes of coefficients, etc. Using infix vs prefix
notation is also analyzed. The number of variables is either 1 or 2,
which seems insufficient. Already PUBK shows that going from 2 to 3
(poly 5 vs poly6) variables reduces the performance considerably.

The main difference to PUBK is that the unnormalized polynomials are
generated in a simpler format (sum of products of factors) that makes
(or should make) the normalization procedure very simple. And that the
evaluation is done step-wise, similar e.g. to work of Gauthier [3].

The setting is a bit problematic when compared to the full setting
also by forcing the normalization to be done in a particular "obvious"
way. A richer set of rewriting steps consisting e.g. of finding common
factors (as e.g. in 2*(2x+1)*(y+1)*(z+2) + 2*(2x+1)*(y+1)*(z+3)) could
lead to shorter rewriting sequences. These two issues - very
simple polynomials and very constrained rewriting rules - imply that
the setting is insufficient to answer the question posed by the title.

My overall feeling is that the paper shows a lot of experimental data,
but it does not bring sufficiently interesting new insights.

Some more comments:

The poor generalization (e.g. Table 16) of the transformer to symbolic
data generated differently is not very surprising. Still, I am missing
information about testing transformers trained on fewer variables on
data with more variables.

On page 5 the authors say "We make sure that the simplified versions
of the input polynomial in the training batches, do not collide with
any endpoints in the  test and validation set."
==>
How was this done? PUBK shows that one simple kind of replacement in data (CONST
instead of all digits) leads to very large train/test overlaps in
LC. But PUBK says that this is initial and much more needs to be
done. A simple improvement suggested there is measuring the
Levenshtein distance as in Wang et al., 2018. [1]. I would expect much
more on this topic given the previous work and their issues.

Compared with Zombori et al. [2], none of the settings is ever
shown to learn perfectly a usable (even if simple) algorithm.

p2: "As a state-of-the-art model, we explore Transformers. While both Graph Neural Networks and
Transformers have been used for single-step representation learning of symbolic theorems and single
step goal-theorem scoring, Transformer-based sequence-to-sequence networks have shown superior-
ity in end-to-end tasks in integration, differential equations"
==>
Various versions of tree neural nets have been used quite successfully by Gauthier for related symbolic tasks [3].  Similarly for guiding theorem provers, in particular in ENIGMA [4] .

Prefix vs infix: see [5] for previous related work on this and more.

p2: symbolic re-write==> symbolic rewriting

p2: the facstep in the example is unclear/confusing - X^1 is replaced just by X.
==> What is the underlying representation? Can you give a more illustrative example of facstep?


References:

[1] Qingxiang Wang, Cezary Kaliszyk, Josef Urban:
First Experiments with Neural Translation of Informal to Formal Mathematics. CICM 2018: 255-270

[2] Zsolt Zombori, Adrián Csiszárik, Henryk Michalewski, Cezary Kaliszyk, Josef Urban:
Towards Finding Longer Proofs. CoRR abs/1905.13100 (2019)

[3] Thibault Gauthier:
Deep Reinforcement Learning for Synthesizing Functions in Higher-Order Logic. LPAR 2020: 230-248

[4] Karel Chvalovský, Jan Jakubuv, Martin Suda, Josef Urban:
ENIGMA-NG: Efficient Neural and Gradient-Boosted Inference Guidance for E. CADE 2019: 197-215

[5] Bartosz Piotrowski, Josef Urban:
Stateful Premise Selection by Recurrent Neural Networks. LPAR 2020: 409-422

========================

UPDATE

The response says:

"With the straightforward use of Transformers, where the model has only seen a single variable in training, there’s no information for the model about what to do with the second variable and thus it will not generalize to the two variable case. Training on two variable-polynomials and testing on two variable-polynomials has relatively low accuracies in our experiment. This suggests that training on single variable-polynomials and testing on two variable-polynomials will result in even lower accuracies. With more work, one may be able to design a model with appropriate inductive bias that understands the concept of multiple variables. This is beyond our scope."

I am afraid that this makes the study rather insufficient for me. The problem of representing variables, eigenvariables/skolems, and capturing structural similarity between different theories and signatures is ubiquitous in the ML-for-TP area. Practically all useful systems developed so far - both features-based and DL-based - have to address this. The authors' answer is "our representation is unsuitable". The observation that if you have no shared representation of variables, you will get little/no generalization is a no-brainer and there is hardly any need to publish negative papers about it. In particular, in a conference about *representations* and some 15 years after first useful systems dealing with such issues have been developed. There are many fixes to this - see e.g. Gauthier's representation of variables in his Tree NNs, etc.

My score will stand, but I would like to encourage the authors to dig deeper and follow the suggestions given in this and other reviews. The general topic of learnability of symbolic rewriting by various neural architectures is certainly interesting, potentially very useful, and far from well understood.

---

> ### Author Response · Authors · 2020-11-19
> **Response to AnonReviewer3**
>
> Dear Reviewer, we appreciate your detailed comments on the work. We put your comments in quotes, followed by our responses.
> “The number of variables is either 1 or 2”:  This relates to the main aim of the work. The main aim is to test Transformers on the polynomial simplification task  in a fine-grained manner; i.e. from step-wise understanding all the way down to basic operator level understanding. While it is true that we could have looked at more variables, our results for two variables already show considerable decrease in accuracy.
>
> Claimed similarity with Gauthier [3]: Indeed, many of the works on neural theorem provers cited in the paper produce stepwise proofs and Gauthier [3] appears to be in the same category. As explained in the paper, our setting is distinct from these in that no search is involved and thus the task is much simpler.
>
> “These two issues - very simple polynomials and very constrained rewriting rules - imply that the setting is insufficient to answer the question posed by the title.”: We agree that the precise task that we study is a special type of polynomial simplification. However, we think the above comment is misplaced for several reasons:
> (1) While the title of the paper does mention polynomial simplification without qualification, the task is very clearly specified in the paper from the beginning and there's no claim that we are studying *the* polynomial simplification task. To our knowledge, there's no *canonical* polynomial simplification task. Indeed, there’s no limit to how complex one can allow the rewrite steps to be. Given this, we think that the title is descriptive and not unreasonable as it is difficult to specify the details in the title.
> (2) Our negative results show that Transformers fail even for such a simple task in some of the settings (e.g., two variables results in Table 2). This strongly suggests that Transformers will fail if we increase the complexity of the task, e.g. by allowing richer rewrite rules or increasing the number of variables. For negative results, failure on a simpler task is a *stronger* result.
> (3) Allowing richer rewriting rules is interesting in its own right but not relevant for our purpose here which is to study what Transformers allow us to do in a very simple setting. Forcing the normalization to be done in the unique (or “obvious”) way is an important design feature of our study that enables us to do this. We can of course give up on uniqueness and generate a dataset without unique proofs, and increase the complexity of the task in other ways. We suspect that that will further degrade the performance of Transformers.
> “it does not bring sufficiently interesting new insights”: Please see our general response.
>
> “Still, I am missing information about testing transformers trained on fewer variables on data with more variables.”: With the straightforward use of Transformers, where the model has only seen a single variable in training, there’s no information for the model about what to do with the second variable and thus it will not generalize to the two variable case. Training on two variable-polynomials and testing on two variable-polynomials has relatively low accuracies in our experiment. This suggests that training on single variable-polynomials and testing on two variable-polynomials will result in even lower accuracies. With more work, one may be able to design a model with appropriate inductive bias that understands the concept of multiple variables. This is beyond our scope.
>
> Train Test Disjoint: We ensure that no unnormalized polynomial in train set simplifies to the same polynomial and one in the test set. This ensures that none of the intermediate steps could also be the same.
>
> CONST substitution by PUBK: The CONST substitution done by PUBK [1] on the LC [2] dataset is not applicable for the polynomial dataset as here arithmetic is an integral part of our task, which is not the case for integration.
>
> “Compared with Zombori et al. [2], none of the settings is ever shown to learn perfectly a usable (even if simple) algorithm”: This comment too seems to stem from a misunderstanding of our goals. We are not sure which result in [2] is being referred to here.
>
>  “What is the underlying representation? Can you give a more illustrative example of facstep?”
> The underlying representation is simply the inorder (infix) or preorder (prefix) traversal of the expression tree. So, 2 * x1 ^ 2 * x2 is written as [2, MUL, x1, EXP, 2, MUL, x2] in infix.
> Facstep example: 2 * x1 ^ 1  + 1 * x2 -> 2 * x1 + x2
> Representation: [2, MUL, x1, EXP, 1, ADD, 1, MUL, x2] -> [2, MUL, x1, ADD, x2]

---

### Official Review · AnonReviewer1 · 2020-10-29
**A somewhat interesting set of experiments with unclear takeaways.**

**Rating:** 4
**Confidence:** 4

**Review:**

This paper studies the efficacy of transformers on a polynomial simplification tasks.

There are two main motivations for this work: Piotrowski et al and, Lample and Sarton (references in the paper). The paper is set out to explore the capability of transformer networks of creating muti-step proofs.

One of the contributions of the paper is the creation of dataset of polynomial simplifications. They use the method in Lample and Charleston to generate a large random dataset of polynomials represented as a sum of products. Each term in that product is a product of a small set of factors. So the basic question is: how do we represent this polynomial by a formula of minimum length, in which each operation is one of the +, - or *.

This is a hard question for sure.

But is this question as hard as in Lample and Sarton?

Is it hard as computing integrals of expressions?

Everybofy knows that computing integrals is hard. In fact, it is much harder than computing partial derivatives.

How hard is it to give a representation of a sum of products? It might be hard. But is it as hard as the above?

I can't tell, but this paper does not even specify what is a baseline. When do we believe that something is important.

This paper fails to specify what is an interesting message and fails to specify that message.

---

> ### Author Response · Authors · 2020-11-19
> **Response to AnonReviewer1**
>
> Thank you for your review. Please see our general response, as most of the objections/concerns are related to the basic premise of the work. In short, our aim is not to test on harder problems, but to test the transformer's ability to do multi-step reasoning. Within the scope of a somewhat simpler setting of polynomial simplification tasks, our results map out the strengths and weaknesses of Transformers.

---

> > ### Comment · AnonReviewer1 · 2020-11-19
> > **Takeaway is still unclear**
> >
> > You claim that the goal is to study multi-step reasoning capabilities as opposed to [Lample and Charton].
> >
> > That's a justifiable difference. On the other hand I am still kind of lost regarding the takeaways of the paper.
> >
> > The paper addresses a task that can be relatively efficiently handled by other methods and there has been overwhelming evidence for Transformer's capability to perform one and multi-step reasoning tasks. ([Polu and Sutskever: GPT-f], [Lample and Charton], [Rabe et al Skip-tree training], [Wu et al: INT] etc.)
> >
> > This paper might add another measurement point to this but it is neither very novel nor very interpretable.
> >
> > My main problem is that if I would need to describe what I have learned from this paper, I had a hard time as it is really hard to put it in a context in which it has a few clear takeaways.
> >
> > If it is the hardness of integer multiplication, then that would have been best studied in isolation.
> > If it is the usefulness of curriculum learning, then we have more impressive measurement points: [Zombori et al], also the paper does not describe a generally usable strategy to create curricula.
> > If it is the good performance on this particular simplification task, then it is unclear how it compares with existing baseline methods.
> >
> > In general, I find this paper not systematic enough and light on clear practical results and takeaways.

---

> > > ### Author Response · Authors · 2020-11-20
> > > **The Takeaway**
> > >
> > > Dear Reviewer, Thank you for engaging with us and your thoughtful reply.
> > >
> > > “The paper addresses a task that can be relatively efficiently handled by other methods and there has been overwhelming evidence for Transformer's capability to perform one and multi-step reasoning tasks. ([Polu and Sutskever: GPT-f], [Lample and Charton], [Rabe et al Skip-tree training], [Wu et al: INT] etc.)”
> > >
> > > Our paper is precisely calling into question what you term “overwhelming evidence” from some of the papers you cite and others. As we argued in the paper and our response, many of these deal with more complex types of reasoning than we do. Given that Transformers are unable to do two variable polynomial simplification well (within our setting), it raises concerns about what they are doing in those other settings where test errors are higher and the possibility of probing what’s going on far more limited because of the complexity of the task and the human written datasets. Our work is only a first step in this direction and we believe further synthetic settings can be devised that test other aspects of reasoning in a systematic way.
> > >
> > > You are right that there are multiple observations in the paper and none stands out. In our view, this is not a shortcoming but inherent in the nature of the problem: reasoning task is multifaceted where one necessarily has to deal with multiple phenomena and there are multiple failure modes. It is of course possible to study single phenomena (e.g. integer multiplication) which can be interesting in its own right, but that’s a different problem and both types of studies are valuable. We found that to shoehorn the conclusions of such a study into one or two major technical conclusions is not possible. However, a high level one-line summary can be given: The situation with Transformer’s mathematical reasoning abilities is muddy.
> > >
> > > “If it is the usefulness of curriculum learning, then we have more impressive measurement points: [Zombori et al], also the paper does not describe a generally usable strategy to create curricula.”
> > >
> > > The two works cannot be directly compared due to different settings (they work with more complex theorem proving problems) and different models (they use reinforcement learning).
> > >
> > > "The Takeaway is still not clear"
> > > As a recount, we analyze multi-step reasoning capabilities of Transformers in a systematic and careful way spanning various dimensions. Our insights *together summarizes* the strengths and weaknesses that Transformers exhibit. Additionally, we also believe that the proposed style of analysis and derived insights will motivate the community to  *analyze across the discussed dimensions for any task that require multiple-step reasoning*. As we show from or results, we believe this is necessary before claiming such tasks are solved. Currently we have seen that, the end-to-end proof accuracy for integration/DEs (in LC), HOList (Bansal et al.) etc. or next-step prediction accuracy analysis do not capture such dimensions.

---

> > > > ### Comment · AnonReviewer1 · 2020-11-20
> > > > **Is Muddiness a takeway?**
> > > >
> > > > Thanks a lot for the thorough answer!
> > > >
> > > > I agree that [Zombori et al] is not directly comparable to your work, as it has a different context and task. Still the fact that they could learn to perform proofs with 100s or even over 1000 steps was a very convincing demonstration of the power of a thoughtfully engineered curriculum. It is true that that work required the curriculum even more because it was in an RL setting, while this work is more direct. Still the news that a carefully engineered training-schedule can become critical is not very novel or surprising anymore (there are other search problems when similar conclusions are imminent.) However,  this paper offers little new in this respect besides the usefulness of some ad hoc curriculum for a somewhat ad hoc and little studied, artificial problem.
> > > >
> > > > The other message is that transformers can't learn certain tasks easily. Unfortunately such negative lessons are not super useful on their own, unless supported by overwhelming evidence and interesting questions. For example, it is surprising how relatively sophisticated segmentation models do poorly when trained on highly synthetic segmentation tasks and unlimited data. The same model might perform very well on the coco and imagenet benchmarks. So failure on highly synthetic tasks be a cause of concern but is not necessarily a show-stopper or super interesting, publishable insight (the above observation is also unpublished, but shows that we encounter a lot of failures, and without a very thorough and exhaustive search and validation, those negative results tend to be highly suspect, since there can be a lot of major bugs, suboptimal hyperparamaters, or other unexpected issues that can affect the validity of the results. This means that negative results should always be accompanied by extreme amount of evidence and also a clear qualitative description of the exact failure modes together with explicit attempts of trying to fix them.
> > > >
> > > > I still feel that after reading this paper, if someone would ask me next week about it, I would really have a hard time describing what I have learned from it, exactly. And even the supposed takeaway is kind of doubtful, since the experiments do not seem to be sufficiently exhaustive or systematic.

---

> > > > > ### Author Response · Authors · 2020-11-23
> > > > > **Follow-up**
> > > > >
> > > > > Thank you for your reply. We will respond to the new points that you raise.
> > > > >
> > > > > "So failure on highly synthetic tasks be a cause of concern but is not necessarily a show-stopper or super interesting, publishable insight"
> > > > >
> > > > > Rather than stopping the show, the intention is to examine more closely what has been achieved. Related to the example you mention, it's well-known that computer vision models are susceptible to adversarial attacks and often learn spurious features. These things can be demonstrated by artificial examples. Arguably, a large fraction of adversarial examples are artificial as they don’t arise in natural settings. Synthetic artificial settings let one control the number of moving parts and simplify the problem.
> > > > >
> > > > > "those negative results tend to be highly suspect, since there can be a lot of major bugs, suboptimal hyperparamaters, or other unexpected issues that can affect the validity of the results. This means that negative results should always be accompanied by extreme amount of evidence and also a clear qualitative description of the exact failure modes together with explicit attempts of trying to fix them."
> > > > >
> > > > > Completely agreed. For the medium coefficient, prefix input, un-annotated proofs, the experiments were:
> > > > > 1. learning rates: 1e-2, 1e-3, 1e-4, 5e-5 with embedding size 256. With 1e-2, loss did not seem to decrease steadily. 1e-3 resulted in low accuracy. 1e-4 worked well and 5e-5 had slow convergence.
> > > > > 2. Setting learning rate as 1e-4, tried different embedding sizes: 128, 256, 512.
> > > > >     Accuracy wise, both 256 and 512 fared equally well and better than 128.
> > > > > 3. With learning rate 1e-4, embedding size 256, tried dropout as 0 and 0.5. We got similar accuracy for both.
> > > > > All other experiments were run with lr 1e-4, embedding size 256, dropout 0.
> > > > > We think the extensive tables and discussion in the paper are pretty explicit description of failure modes.

---

### Author Response · Authors · 2020-11-19
**General Response**

Previous work in symbolic math tasks such as PUBK [1] and LC [2] focus on the end-to-end setting, and get quite good performance. Previous work on logic tasks such as HOL [3], INT [4] focus on stepwise proof search, and achieve moderately good performance. However, generating stepwise proofs on symbolic tasks, which is interesting in its own right, isn’t as well explored.
In this paper, we propose a very simple mathematical setting to evaluate Transformers' ability in generating proofs on the symbolic task of polynomial simplification. Thus, our goal is not to give new state-of-the-art models nor is our goal to extract a usable algorithm from the model. Our results map out, within the scope of our polynomial simplification problem, the strengths and weaknesses of Transformers. What we lose by making the task simple, we gain by being able to study the problem more extensively and probing the model along multiple axes. This can inform the future work on designing better models and training algorithms:

(1) As a concrete example, we identify integer multiplication as one of the main bottlenecks. In general, neural nets struggle with integer multiplication. One way to make progress then could be to design a neuro-symbolic system where the multiplication part is performed by a symbolic algorithm. More generally, decomposing the task into components where the model works well and those where it does not can help design better systems.

(2) As another example, we see that for some settings, curriculum learning results in significant benefits.

(3) One general finding is that accuracy at end-point tasks is often significantly better than proof accuracy. Our models do not have a notion of proof, they only see individual proof steps. Thus incorporating the notion of proof in the model, perhaps by modifying the loss function to take the proof into account might help.

[1] Bartosz Piotrowski, Josef Urban, Chad E. Brown, and Cezary Kaliszyk. Can neural networks learn symbolic rewriting?, 2019.

[2] Guillaume Lample and Francois Charton. Deep learning for symbolic mathematics. In International Conference on Learning Representations, 2020.

[3] Aditya Paliwal, Sarah M. Loos, Markus N. Rabe, Kshitij Bansal, and Christian Szegedy. Graph representations for higher-order logic and theorem proving. In The Thirty-Fourth AAAI Conference on Artificial Intelligence, AAAI 2020, The Thirty-Second Innovative Applications of Artificial Intelligence Conference, IAAI 2020, The Tenth AAAI Symposium on Educational Advances in Artificial Intelligence, EAAI 2020, New York, NY, USA, February 7-12, 2020, pp. 2967–2974. AAAI Press, 2020. URL https://aaai.org/ojs/index.php/AAAI/ article/view/5689.

[4] Yuhuai Wu, Albert Jiang, Jimmy Ba, and Roger B. Grosse. INT: an inequality benchmark for evaluating generalization in theorem proving. CoRR, abs/2007.02924, 2020. URL https://arxiv.org/abs/2007.02924.

---

### Decision · Program_Chairs · 2021-01-07
**Final Decision**

**Decision:**

Reject

**Comment:**

While the reviewers find the experiments in the paper somewhat interesting, they find that the paper does not sufficiently address whether the limitations shown for models in this paper translate to larger models and other, more realistic, tasks, or an artifact of the setup considered in the paper.  Overall the takeaways seem unclear from the paper and I believe it is not ready for acceptance.  Addressing the issues raised by reviewers and having a more clear discussion on connections to existing results will help the paper.